# Learning in the Presence of Low-dimensional Structure: A Spiked Random Matrix Perspective

**Jimmy Ba**[1,2,3], **Murat A. Erdogdu**[1,2], **Taiji Suzuki**[4,5], **Zhichao Wang**[6], **Denny Wu**[7,8]

[1]University of Toronto, [2]Vector Institute, [3]xAI, [4]University of Tokyo, [5]RIKEN AIP,
[6]University of California San Diego, [7]New York University, [8]Flatiron Institute

{jba,erdogdu}@cs.toronto.edu, taiji@mist.i.u-tokyo.ac.jp,
zhw036@ucsd.edu, dennywu@nyu.edu

## Abstract

We consider the problem of learning a single-index target function $f_* : \mathbb{R}^d \to \mathbb{R}$ under the spiked covariance data:

$$f_*(\boldsymbol{x}) = \sigma_*\left(\tfrac{1}{\sqrt{1+\theta}}\langle \boldsymbol{x}, \boldsymbol{\mu}\rangle\right), \quad \boldsymbol{x} \sim \mathcal{N}(0, \boldsymbol{I}_d + \theta\boldsymbol{\mu}\boldsymbol{\mu}^\top), \quad \theta \asymp d^\beta \text{ for } \beta \in [0, 1),$$

where the link function $\sigma_* : \mathbb{R} \to \mathbb{R}$ is a degree-$p$ polynomial with information exponent $k$ (defined as the lowest degree in the Hermite expansion of $\sigma_*$), and it depends on the projection of input $\boldsymbol{x}$ onto the spike (signal) direction $\boldsymbol{\mu} \in \mathbb{R}^d$. In the proportional asymptotic limit where the number of training examples $n$ and the dimensionality $d$ jointly diverge: $n, d \to \infty, n/d \to \psi \in (0, \infty)$, we ask the following question: how large should the spike magnitude $\theta$ be, in order for $(i)$ kernel methods, $(ii)$ neural networks optimized by gradient descent, to learn $f_*$? We show that for kernel ridge regression, $\beta \geq 1 - \frac{1}{p}$ is both sufficient and necessary. Whereas for two-layer neural networks trained with gradient descent, $\beta > 1 - \frac{1}{k}$ suffices. Our results demonstrate that both kernel methods and neural networks benefit from low-dimensional structures in the data. Further, since $k \leq p$ by definition, neural networks can adapt to such structures more effectively.

## 1 Introduction

**Learning under spiked covariance.** Real-world data is often high-dimensional but it also exhibits certain low-dimensional structures. Indeed, the number of input features is exceedingly large in machine learning, yet most relevant information is concentrated in a low-dimensional subspace [LV07, HTFF09]. Such structures have important consequences on learning performance. In random matrix theory, high dimensionality is reflected by the *proportional asymptotic limit* where the numbers of training examples and input features diverge to infinity at the same rate, whereas the low-dimensional structure is often described by a *spiked random matrix model* [Joh01, BAP05, BS06, BGN11] in which a low-rank signal ("spike") is hidden in a large-dimensional noise ("bulk"). In this work, we restrict ourselves to the simple setting where the signal is rank-1, and consider the following data-generating process:

$$\boldsymbol{x} \sim \mathcal{N}(0, \boldsymbol{I}_d + \theta\boldsymbol{\mu}\boldsymbol{\mu}^\top), \quad f_*(\boldsymbol{x}) = \sigma_*\left(\tfrac{1}{\sqrt{1+\theta}}\langle \boldsymbol{x}, \boldsymbol{\mu}\rangle\right), \tag{1.1}$$

where the goal is to estimate the target function (teacher) $f_*$, which is a *single-index model* depending on the signal direction $\boldsymbol{\mu} \in \mathbb{R}^d$, and $\sigma_* : \mathbb{R} \to \mathbb{R}$ is an unknown *link function*. The signal $\boldsymbol{\mu}$ is also embedded in the spiked input data, and $\theta > 0$ controls the spike magnitude; we scale $\theta \asymp d^\beta$ for $\beta \in [0, 1)$, where larger $\beta$ indicates a more prominent low-dimensional structure, and hence the learning problem becomes easier. We aim to characterize the efficiency of learning $f_*$ using kernel methods and neural networks (NNs) in relation to the strength of the low-dimensional signal $\theta$.

37th Conference on Neural Information Processing Systems (NeurIPS 2023).

**Prior results: isotropic data.** When $\theta = 0$, the input features are isotropic and do not reveal any information about the target $f_*$. In this setting, the performance of kernel methods and two-layer NNs have been extensively studied in the proportional asymptotic limit.

- **Kernel methods.** For isotropic Gaussian or spherical input data, the performance of random features (RF) regression and kernel ridge regression (KRR) has been precisely characterized in the proportional limit [MM22, GLK+20, DL20, BMR21]. However, in this regime, RF models and KRR suffers from the "curse of dimensionality" [Bac17, YS19] and cannot achieve vanishing generalization error unless $f_*$ is *linear* [EK10, GMMM21, HL22]. More generally, for KRR to learn a degree-$p$ polynomial target, a sample complexity of $n = \Omega(d^p)$ is required [GMMM21, LRZ20, DWY21, MMM21].

- **Neural networks + Gradient descent.** While NNs can efficiently *approximate* a single-index model (with polynomial link $\sigma_*$), the *optimization* complexity of gradient-based learning on isotropic data is known to be governed by the *information exponent*, defined as the index of the first non-zero Hermite coefficient of $\sigma_*$ [AGJ21]. For NNs trained with gradient descent (GD) to learn a target with information exponent $k$, recent works showed that a sample complexity of $n = \tilde{\Theta}(d^k)$ suffices[1] [AGJ21, DLS22, BBSS22, ABAM23]. Hence, in the proportional regime ($n \asymp d$), learnability has only been established for $k = 1$ [AAM22, BES+22, MHPG+23, BMZ23].

These prior results delineate the limitation of kernel methods and GD-trained NNs under isotropic data in the high-dimensional proportional scaling: KRR can only learn linear functions on the input, and NNs learn targets with information exponent $k = 1$, which cannot cover many important problems such as phase retrieval [Fie82, CC15, TV23]. Following the intuition that low-dimensional structure in the input data can make the learning problem easier, we ask the following question.

> *If the input data contains low-dimensional structure given by the spiked covariance model* (1.1), *can kernel methods and neural networks learn a larger class of $f_*$ in the proportional regime?*

## 1.1 Our Contributions

We answer this question in the affirmative by showing that in the proportional asymptotic limit when the spike magnitude $\theta$ reaches a certain threshold, both KRR and GD-trained two-layer NN can learn the target function $f_*$, where the link function $\sigma_*$ is a degree-$p$ polynomial with information exponent $k$ (see Definition 1). Our findings are summarized below and illustrated in Figure 1.

- For KRR with the inner-product kernel, we give a sharp analysis of the prediction risk and show that to learn a link function $\sigma_*$ with degree $p$ in the proportional regime, a spike magnitude of $\theta = \Omega(d^{1-\frac{1}{p}})$ is *both necessary and sufficient*.

- For a two-layer NN with ReLU activation, we upper bound the prediction risk when representation learning is performed via one gradient descent step on the first-layer parameters (analogous to [BES+22, DLS22]), and show that a spike magnitude of $\theta = \omega(d^{1-\frac{1}{k}})$ is *sufficient* to learn $\sigma_*$ with information exponent $k$.

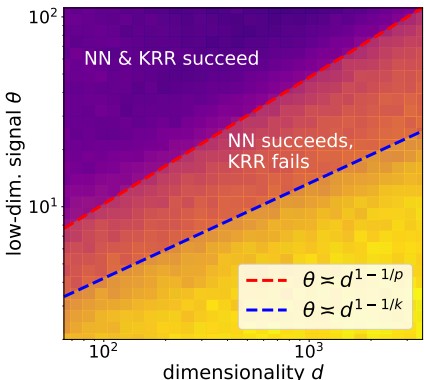

Figure 1: Spike magnitude $\theta \asymp d^\beta$ for KRR and GD-trained NN to learn the target $f_*$ with degree $p$ and information exponent $k$ in the proportional regime. When $k \neq p$, KRR requires a stronger low-dimensional structure (i.e., larger spike) to learn $f_*$.

Our analysis reveals that both KRR and NNs can benefit from the low-dimensional structure (spike) in the input data, and they can learn a wider class of target functions in the proportional regime compared to the isotropic setting (i.e., $p > 1$ for KRR, and $k > 1$ for NNs). Furthermore, since for a given link function $k \leq p$ by definition, we know that NNs require a smaller spike magnitude $\theta$ to learn the same target compared to KRR (see Figure 1). In other words, a GD-trained NN can utilize such a low-dimensional structure more effectively than KRR.

---

[1]The appearance of $k$ in the exponent of the dimensionality is also consistent with the correlational statistical query (CSQ) lower bound [DLS22, ABAM23], which relates to the hardness of learning via gradient descent.

**Related works.** Related to our problem setting, [GMMM20] studied the sample complexity of KRR and NNs in a spiked covariance model, and showed that both models indeed benefit from a stronger low-dimensional structure (i.e., larger spike). However, their analysis assumed the number of spikes to be diverging and cannot cover the proportional regime we consider. More importantly, the NN analysis is limited to approximation, and gradient-based optimization guarantee is not given.

In [RGKZ21], the authors presented a non-rigorous analysis on classifying XOR mixtures using an RF model and a two-layer NN trained with online SGD and showed that the NN can learn the task under smaller spike magnitude. Such a task is parallel to the information exponent $k = 2$ setting (see [ABAM23]); our results rigorously demonstrate that similar separation exists for a different class of target functions with potentially higher degree and information exponent.

## 2 Preliminaries: Problem Setting and Assumptions

**Notations.** $\| \cdot \|$ denotes the $\ell_2$ norm for vectors and the $\ell_2 \to \ell_2$ operator norm for matrices, and $\| \cdot \|_F$ is the Frobenius norm. $\mathcal{O}_d(\cdot)$ and $o_d(\cdot)$ stand for the standard big-O and little-o notations, where the subscript highlights the asymptotic variable; we write $\tilde{\mathcal{O}}(\cdot)$ when (poly-)logarithmic factors are ignored. $\mathcal{O}_{d,\mathbb{P}}(\cdot)$ (resp. $o_{d,\mathbb{P}}(\cdot)$) represents big-O (resp. little-o) in probability as $d \to \infty$. $\Omega(\cdot), \Theta(\cdot), \omega(\cdot)$ are defined analogously. Denote $\gamma$ as the standard Gaussian distribution in $\mathbb{R}$. Given $f : \mathbb{R}^d \to \mathbb{R}$, we denote its $L^p$-norm with respect to the data distribution as $\|f\|_{L^p}$.

### 2.1 Basic Assumptions

We consider Gaussian input data with a spiked covariance:

$$\boldsymbol{x} \sim \mathcal{N}(0, \boldsymbol{I}_d + \theta \boldsymbol{\mu}\boldsymbol{\mu}^\top), \quad \theta \asymp d^\beta \text{ for some } \beta \in [0, 1), \tag{2.1}$$

where the signal direction $\boldsymbol{\mu} \in \mathbb{R}^d$ satisfies $\|\boldsymbol{\mu}\| = 1$, and the spike magnitude $\theta$ is allowed to grow with the input dimensionality $d$ as specified by the exponent $\beta$: as $\beta$ gets larger, the spike magnitude increases and hence the data exhibits stronger low-dimensional structure. Following the terminology in [GMMM20], we may define the *effective dimensionality* of the input data as $d_{\text{eff}} \asymp d^{1-\beta}$, which captures the intuition that a larger spike renders the features more low-dimensional. As we will see, the sample complexity of KRR and two-layer NN is decided by this effective dimensionality.

**Teacher model.** We consider a *student-teaching setting*, where the labels $y$ are generated from a teacher model (target function) $f_* : \mathbb{R}^d \to \mathbb{R}$. In the spiked covariance model (2.1), it is known that input directions with large variations are often good predictors of the labels — indeed, this is the main reason that principal component analysis is used in practice [HTFF09]. We therefore consider the following single-index target function where the index features align with the spike direction $\boldsymbol{\mu}$:

$$y = f_*(\boldsymbol{x}) = \sigma_* \left( \tfrac{1}{\sqrt{1+\theta}} \langle \boldsymbol{x}, \boldsymbol{\mu} \rangle \right), \tag{2.2}$$

where the link function $\sigma_* \in L^2(\gamma)$ is centered such that $\mathbb{E}[\sigma_*(z)] = 0$ for $z \sim \mathcal{N}(0, 1)$ (this can be achieved by subtracting the mean from the training labels as in [DLS22]), and the normalization factor $(1 + \theta)^{-1/2}$ ensures that the $L^2$ norm of $f_*$ remains constant. We remark that given the prior knowledge of low-dimensional structure, we may efficiently learn $f_*$ by first performing PCA on the input features; our goal, however, is not to construct an optimal learning algorithm, but to understand the behavior of KRR and two-layer NN without such data preprocessing, similar to [GMMM20].

We also assume $\sigma_*$ is a degree-$p$ polynomial with information exponent (IE) $k$ defined below.

**Definition 1** (Information exponent). *Let $\{h_j\}_{j=0}^\infty$ denote the normalized Hermite polynomials. The information exponent of $f \in L^2(\gamma)$, which we denote by $k$, is the index of the first non-zero Hermite coefficient of $f$, that is, given the Hermite expansion $f = \sum_{j=0}^\infty \alpha_j h_j$, $k := \min\{j > 0 : \alpha_j \neq 0\}$.*

It is clear that by definition, we always have $k \leq p$, and equality is only achieved when $\sigma_*$ is a "pure" degree-$k$ Hermite polynomial. Intuitively, IE measures the magnitude of information contained in the gradient, and larger $k$ implies increased gradient descent complexity [AGJ21, BBSS22]. Such definition is also related to the recently introduced *leap complexity* [ABA22].

**Remark.** *In (2.2) we do not corrupt the training labels $y$ with i.i.d. label noise, meaning that, in light of the bias-variance decomposition, we analyze the bias term describing the extent $f_*$ is learned by the student model. If label noise is introduced, the additional variance term (describing the "overfitting" due to noise) can be handled by standard concentration argument.*

## 2.2 Learning Objective and Training Procedure

Given training examples $\{(\boldsymbol{x}_i, y_i)\}_{i=1}^n$, where $\boldsymbol{x}_i \overset{\text{i.i.d.}}{\sim} \mathcal{N}(0, \boldsymbol{I}_d + \theta \boldsymbol{\mu}\boldsymbol{\mu}^\top)$ and $y_i = f_*(\boldsymbol{x}_i)$, we consider two student models obtained via the following training procedure in the proportional regime:

$$\textit{Proportional asymptotic limit: } n, d \to \infty, \quad n/d \to \psi \in (0, \infty). \tag{2.3}$$

**Student model I: Kernel ridge regression.** We focus on the inner-product kernel: $k(\boldsymbol{x}, \boldsymbol{y}) = g\left(\frac{\langle \boldsymbol{x}, \boldsymbol{y} \rangle}{d}\right)$ for some $g \in L^2(\gamma)$ (similar argument can also apply to rotationally invariant kernels as in [EK10, DWY21]). Denoting the associated reproducing kernel Hilbert space (RKHS) with $\mathcal{H}$, the kernel ridge regression estimator is given by

$$\hat{f}_{\text{ker}} = \operatorname*{argmin}_{f \in \mathcal{H}} \left\{ \frac{1}{n} \sum_{i=1}^n (y_i - f(\boldsymbol{x}_i))^2 + \lambda \|f\|_{\mathcal{H}}^2 \right\} \Rightarrow \hat{f}_{\text{ker}}(\boldsymbol{x}) = k(\boldsymbol{x}, \boldsymbol{X})^\top (\boldsymbol{K} + \lambda \boldsymbol{I})^{-1} \boldsymbol{y}, \tag{2.4}$$

where $\lambda > 0$ is the ridge parameter, and $\boldsymbol{K} \in \mathbb{R}^{n \times n}$ is the kernel Gram matrix with entries $\boldsymbol{K}_{i,j} = k(\boldsymbol{x}_i, \boldsymbol{x}_j)$. Spectral properties of the kernel matrix and the prediction risk of KRR have been extensively studied in various high-dimensional settings [EK10, GMMM21, BMR21, XHM+22, LY22]. Since the kernel function is fixed before seeing the data, we intuitively expect that KRR cannot effectively adapt to a low-dimensional structure in the learning problem.

**Student model II: Two-layer neural network.** We aim to learn the following width-$N$ network:

$$f_{\text{NN}}(\boldsymbol{x}) = \frac{1}{\sqrt{N}} \sum_{i=1}^N a_i \sigma(\langle \boldsymbol{x}, \boldsymbol{w}_i \rangle + b_i) = \left\langle \boldsymbol{a}, N^{-1/2} \sigma(\boldsymbol{W}^\top \boldsymbol{x} + \boldsymbol{b}) \right\rangle, \tag{2.5}$$

where $\sigma : \mathbb{R} \to \mathbb{R}$ is a nonlinear activation, and the trainable parameters $\boldsymbol{W} \in \mathbb{R}^{d \times N}, \boldsymbol{a}, \boldsymbol{b} \in \mathbb{R}^N$ are initialized from standard Gaussian distribution. We require the network width to be $N = \Omega(d^\varepsilon)$ for some small $\varepsilon > 0$. Such a choice entails that it is sufficient to have $\Theta(d^{1+\varepsilon})$ total training parameters, which is almost proportional to the sample size $n \asymp d$; in contrast, for KRR we need to store the kernel Gram matrix with $n^2$ entries, which is less computationally efficient at test time.

It is known that NNs can adapt to the learning problem via *representation learning*, in which the trained features encode relevant information of the target function. To realize this advantage, we update the parameters in (2.5) via the two-stage procedure introduced in [BES+22, DLS22, BEG+22], where we first learn the representation by taking one gradient descent step on the first-layer parameters $\boldsymbol{W}$, and then estimate the second-layer parameters $\boldsymbol{a}$ separately (which is a convex problem). This procedure is summarized as follows (see Algorithm 1 in Appendix for more details).

1. *Feature learning for the 1st layer.* We optimize the *representation* of the NN via gradient descent, where the training objective is the empirical squared loss $\mathcal{L}(f) = \frac{1}{n} \sum_{i=1}^n (f(\boldsymbol{x}_i) - y_i)^2$. Specifically, denote the $i$-th column of the initialized weight matrix $\boldsymbol{W}_0$ as $\boldsymbol{w}_i^{(0)} \in \mathbb{R}^d$, and the initialized NN as $f_{\text{NN}}^0$, we take one GD step with learning rate $\eta$ as follows,

$$\boldsymbol{w}_i^{(1)} \leftarrow \boldsymbol{w}_i^{(0)} - \eta \sqrt{N} \cdot \nabla_{\boldsymbol{w}_i^{(0)}} \mathcal{L}\big(f_{\text{NN}}^{(0)}\big).$$

Due to the anisotropic input data, the pre-activation $\langle \boldsymbol{x}, \boldsymbol{w}_i^{(1)} \rangle$ may blow up if the trained parameter $\boldsymbol{w}_i^{(1)}$ aligns with the signal direction $\boldsymbol{\mu}$. To circumvent this issue, we normalize each neuron to have (roughly) unit pre-activation, i.e., for $i \in [N]$, we perform the weight normalization:

$$\boldsymbol{w}_i^{(1)} \leftarrow \boldsymbol{w}_i^{(1)} \cdot \left( \frac{1}{n} \sum_{j=1}^n \left\langle \boldsymbol{x}_j, \boldsymbol{w}_i^{(1)} \right\rangle^2 \right)^{-1/2}.$$

At high level, this resembles the often-used normalization layers in deep learning [IS15, BKH16].

2. *Ridge regression for the 2nd layer.* After obtaining the updated first-layer parameters $\boldsymbol{W}_1 \in \mathbb{R}^{d \times N}$, we optimize the second-layer $\boldsymbol{a}$ by solving the ridge regression objective – this can be done by using GD to solve an $\ell_2$-penalized least squares. To circumvent the dependence between the training data $\boldsymbol{X}$ and the learned $\boldsymbol{W}_1$, we follow [BES+22] and estimate the regression coefficients $\hat{\boldsymbol{a}}$ using *a new set of training data* $\{\tilde{\boldsymbol{x}}_i, \tilde{y}_i\}_{i=1}^n$ with the same sample size $n$.

   Denoting the feature matrix on the fresh training set $\{\tilde{\boldsymbol{X}}, \tilde{\boldsymbol{y}}\}$ as $\boldsymbol{\Phi} := \frac{1}{\sqrt{N}} \sigma(\tilde{\boldsymbol{X}} \boldsymbol{W}_1 + \boldsymbol{b}) \in \mathbb{R}^{n \times N}$,

   the ridge regression estimator can be obtained by: $\hat{\boldsymbol{a}} = \operatorname{argmin}_{\boldsymbol{a}} \left\{ \frac{1}{n} \|\tilde{\boldsymbol{y}} - \boldsymbol{\Phi}\boldsymbol{a}\|^2 + \lambda \|\boldsymbol{a}\|^2 \right\}$.

**Learning in the proportional limit.** Given a target function $f_*$ and a learned (student) model $\hat{f}$, we evaluate the model performance using the prediction risk:

$$\mathcal{R}(\hat{f}) := \mathbb{E}_{\boldsymbol{x}}[(\hat{f}(\boldsymbol{x}) - f_*(\boldsymbol{x}))^2] = \|\hat{f} - f_*\|_{L^2}^2,$$

where the expectation is taken over the (anisotropic) Gaussian data distribution. To benchmark the model performance in the proportional limit (2.3), we introduce the following notion of learnability.

**Definition 2** (Learnability in the proportional regime). *We say an algorithm learns the target function $f_*$ in the proportional regime if, for any (small) constant $\epsilon > 0$, there exists some constant $\psi_* > 0$ such that when $n/d \to \psi \geq \psi_*$, the algorithm constructs $\hat{f}$ using $n$ i.i.d. training examples, and achieves $\mathcal{R}(\hat{f}) = \|\hat{f} - f_*\|_{L^2}^2 \leq \epsilon$ with probability 1 when $n, d \to \infty$ proportionally.*

Based on this definition, in order for a student model $\hat{f}$ (e.g., KRR or two-layer NN) to learn the teacher $f_*$ in the proportional regime, it needs to achieve arbitrarily small prediction risk with high probability, and with *linear sample complexity* (i.e., using $n \asymp d$ training samples).

## 3 Kernel Ridge Regression

For an inner-product kernel, the performance of KRR depends on the behavior of the pairwise inner-product of the training examples $\langle \boldsymbol{x}_i, \boldsymbol{x}_j \rangle$. Intuitively speaking, since the inner-products concentrate in high dimensions, the kernel matrix can be approximated in operator norm via low-degree Taylor expansion, and thus KRR can only learn $f_*$ that is low-degree (e.g., see [DWY21, WZ23]). Introducing a low-dimensional structure to the data is one way to counteract such near-orthogonality in high dimensions. Hence we expect that KRR can achieve better performance when the spike magnitude $\theta$ becomes larger – this intuition will be made rigorous in the following subsections.

### 3.1 Sharp Analysis of the Prediction Risk

In this section, we study the prediction risk of KRR defined in (2.4) with a positive semidefinite kernel given by $k(\boldsymbol{x}_i, \boldsymbol{x}_j) = g(\langle \boldsymbol{x}_i, \boldsymbol{x}_j \rangle/d)$, i.e., $g^{(k)}(0) \geq 0$ for all $k$ (see [SS02, Section 13.1]). Recall that the strength of the low-dimensional component is determined by the spike magnitude $\theta \asymp d^\beta$; in the following, we characterize how the exponent $\beta \in [0, 1)$ affects the performance of KRR.

We denote by $P_{\leq \ell} : L^2 \to L^2$ the orthogonal projector onto the subspace of polynomials of degree at most $\ell$ for any $\ell \in \mathbb{N}$. Similarly, we denote by $P_{>\ell} = \text{Id} - P_{\leq \ell}$ the projector to the orthogonal complement where Id is the identity operator. To illustrate this notion, consider $\ell = 1$, then we have

$$f_*(\boldsymbol{x}) = P_{\leq 1}f_*(\boldsymbol{x}) + P_{>1}f_*(\boldsymbol{x}) = a^* + \langle \boldsymbol{b}^*, \boldsymbol{x} \rangle + P_{>1}f_*(\boldsymbol{x}),$$

where $a^*, \boldsymbol{b}^* = \text{argmin}_{a, \boldsymbol{b}} \mathbb{E}_{\boldsymbol{x}}[(f_*(\boldsymbol{x}) - a - \langle \boldsymbol{b}, \boldsymbol{x} \rangle)^2]$. Note that any polynomial $f_*$ with degree at most $\ell$ satisfies $P_{>\ell}f_* = 0$. Denoting the KRR model as $\hat{f}_{\text{ker}}$, we have the following sharp analysis of the asymptotic prediction risk.

**Theorem 3.** *Given any fixed $\ell \in \mathbb{N}$. Suppose that $g^{(k)}(0) > 0$ for all $k \leq \ell$, and the spike magnitude scales as $\theta \asymp d^\beta$ for $\beta \in \left(1 - \frac{1}{\ell}, 1 - \frac{1}{\ell+1}\right)$. Then as $n, d \to \infty$, $n/d \to \psi \in (0, \infty)$, the prediction risk of the KRR estimator $\hat{f}_{\text{ker}}$ with $\lambda = \mathcal{O}_d(1)$ satisfies the following with probability 1,*

$$\mathcal{R}(\hat{f}_{\text{ker}}) - \|P_{>\ell}f_*\|_{L^2}^2 = o_d(1).$$

As a corollary of Theorem 3, based on Definition 2, we can obtain a sharp threshold of $\beta$ for learning a degree-$p$ polynomial link function in the proportional regime (Definition 2).

**Corollary 4.** *Assume $\sigma_*$ is a degree-$p$ polynomial with $p \geq 1$ defined in (2.2). Under the same assumptions of Theorem 3, if $\beta > 1 - \frac{1}{p}$, then the KRR estimator with $\lambda = \mathcal{O}_d(1)$ learns $f_*$ in the proportional regime, i.e., $\mathcal{R}(\hat{f}_{\text{ker}}) = o_d(1)$ with probability 1 when $n, d \to \infty$ proportionally.*

We make the following remarks on Theorem 3 and Corollary 4.

- At a high level, the above theorem aligns with the conclusion of [GMMM20] which assumes spherical data with a diverging number of spikes. Recall that the *effective dimensionality* of our anisotropic Gaussian data is $d_{\text{eff}} \asymp d^{1-\beta}$, and Theorem 3 implies that to extract the signal direction $\boldsymbol{\mu}$ and learn a degree-$p$ teacher model, a sample size of $n = \omega(d_{\text{eff}}^p)$ is required; note that for $\beta > 0$, this implies an improvement over the isotropic setting where the sample complexity is $n = \omega(d^p)$.

- While we only state the result for inner-product KRR (similar to [LRZ20, BMR21]), we expect similar findings for general rotationally invariant kernels: $k(\boldsymbol{x}_i, \boldsymbol{x}_j) = g(\langle \boldsymbol{x}_i, \boldsymbol{x}_j \rangle, \|\boldsymbol{x}_i\|, \|\boldsymbol{x}_j\|)$, since the norm term has negligible contribution to the learning of a single-index target (2.2). In fact, [DWY21] established a lower bound for rotationally invariant KRR, which, when applied to our setting, gives a necessary sample complexity of $n \asymp d_{\text{eff}}^{\Omega(p)}$. Although this lower bound may not be sharp, it illustrates that the required spike magnitude needs to scale with target degree $p$.

## 3.2 Intuition behind the Analysis

We briefly summarize the two main ingredients for the proof of Theorem 3. As a preliminary step, we can rotate the signal $\boldsymbol{\mu}$ to the $\boldsymbol{e}_1$ direction due to the rotational invariance of Gaussian distribution. After such transformation, we have $([\boldsymbol{x}_i]_1, [\boldsymbol{x}_i]_{2:d}) \overset{\text{i.i.d.}}{\sim} \mathcal{N}(0, 1+\theta) \oplus \mathcal{N}(0, \boldsymbol{I}_{d-1})$ with $\theta = \Theta(d^\beta)$.

**Polynomial approximation of the kernel matrix.** Firstly, we approximate the inner-product KRR with some degree-$L$ polynomial kernel in the form of

$$k(\boldsymbol{x}_i, \boldsymbol{x}_j) = \sum_{k=1}^{L} c_k d^{-k} \langle \boldsymbol{x}_i, \boldsymbol{x}_j \rangle^k.$$

Such approximation can be used to establish a learning lower bound for KRR. For instance in the proportional limit, [EK10] showed that for well-conditioned data, the inner-product kernel can be approximated by the linear kernel, which implies that KRR only learns linear $\sigma_*$ [BMR21]. For general deterministic datasets, similar approximation has been studied in [WZ23] where the "orthogonality" of input data determines the polynomial approximation error. In our setting, the inner product $\frac{1}{d} \boldsymbol{x}_i^\top \boldsymbol{x}_j = \mathcal{O}_d(d^{\beta-1})$ depends on the spike magnitude $\theta \asymp d^\beta$. Consequently, we approximate our inner-product kernel by some polynomial kernel whose degree depends on $\beta$.

**Orthogonal polynomial expansion.** While the approximation in the preceding step simplifies the kernel, such analysis is generally not sharp. To refine the result, we expend the polynomial kernel in the Hermite bases (analogous to the analysis of spherical data in [GMMM21, MMM21]). After rotation, the teacher model (2.2) can be written as $f_*(\boldsymbol{x}) = \sigma_*(z)$, for $z \sim \mathcal{N}(0, 1)$ and $[\boldsymbol{x}]_1 = \sqrt{(1+\theta)}z$ independent with other entries $[\boldsymbol{x}]_i$ for $2 \leq i \leq d$. In light of the kernel trick [Ver18, Section 3.7], we write $k(\boldsymbol{x}_i, \boldsymbol{x}_j) = \langle \Phi(\boldsymbol{x}_i), \Phi(\boldsymbol{x}_j) \rangle$ where the feature vector $\Phi(\boldsymbol{x})$ is composed of Hermite polynomials of $\boldsymbol{x}$ up to some degree. Importantly, we only need to extract the Hermite components with respect to the first entry $[\boldsymbol{x}]_1$ to learn a function $\sigma_*(z)$.

# 4 Two-layer Neural Network

During the feature learning phase of NN training, we intuitively expect the parameters to align with the (rank-1) signal direction which allows NN to overcome the "curse of dimensionality". Indeed, in this section, we show that the first gradient step on the first-layer weights $\boldsymbol{W}$ enables the model to "zoom in" to the low-dimensional structure, given that the spike magnitude $\theta$ is sufficiently large.

## 4.1 Upper Bound on the Prediction Risk

We consider the following standard Gaussian initialization for the two-layer NN (2.5):

$$[\boldsymbol{W}_0]_{ij} \overset{\text{i.i.d.}}{\sim} \mathcal{N}(0, 1/d), \quad [\boldsymbol{a}_0]_i \overset{\text{i.i.d.}}{\sim} \mathcal{N}(0, 1/N), \quad [\boldsymbol{b}_0]_i \overset{\text{i.i.d.}}{\sim} \mathcal{N}(0, 1). \tag{4.1}$$

Denote the NN optimized by the two-stage procedure outlined in Section 2.2 as $\hat{f}_{\text{NN}}$. In the sequel, we restrict ourselves to $\sigma(z) = \text{ReLU}(z) = \max\{z, 0\}$. We expect similar characterization to hold when the student activation $\sigma$ has non-zero Hermite coefficients up to degree-$p$ (see [ABAM23]).

The following theorem gives a sufficient condition on the spike magnitude $\theta \asymp d^\beta$ in order for $\hat{f}_{\text{NN}}$ to learn the target function $f_*$ with linear sample complexity (in terms of dimension dependence).

**Theorem 5.** *Consider the NN training procedure in Section 2.2 with Gaussian initialization defined in (4.1), $N = \Omega(d^\varepsilon)$, $\eta = \Omega(N^{1/2+\varepsilon})$, and appropriately chosen $\ell_2$ regularization $d^{\varepsilon-1} \lesssim \lambda \lesssim d^{-\varepsilon}$ for small $\varepsilon > 0$. Then, for $\sigma = ReLU$, we have*

$$\mathcal{R}(\hat{f}_{\text{NN}}) = o_d(1) \text{ with probability 1, when } \beta > 1 - 1/k,$$

*as $n, d \to \infty, n/d \to \psi$. That is, the two-layer ReLU NN can learn the target function with the information exponent $k$ in the proportional regime.*

The above theorem predicts that a spike magnitude of $\theta = \omega(d^{1-\frac{1}{k}})$ is sufficient for a GD-trained two-layer NN to learn $f_*$ in the $n \asymp d$ regime. Similar to prior works [AGJ21, BBSS22], the information exponent $k$ also determines the complexity of the learning problem in our setting: larger $k$ implies a more "difficult" task for GD, hence we need a larger $\theta$ (i.e., stronger low-dimensional signal) to achieve linear sample complexity. Moreover, in light of the *effective dimensionality* defined in [GMMM20], Theorem 5 implies a sample size of $n = \omega(d_{\text{eff}}^k)$ suffices to learn a single-index target with information exponent $k$; this contrasts the $n = \omega(d_{\text{eff}}^p)$ complexity of KRR given in Theorem 3. Such discrepancy will be highlighted and empirically validated in Section 5.

## 4.2 Intuition behind the Analysis

Theorem 5 is established in two steps: first, we prove that under certain conditions, gradient descent can align the first-layer parameters $W$ with the signal direction $\mu$; then we show that after achieving such an alignment, $f_*$ can be efficiently learned by optimizing the second-layer $a$.

**Spiked covariance amplifies gradient signal.** Due to our "mean-field" initialization (i.e., small second-layer coefficients), the initial NN has small output $f_{\text{NN}}^{(0)}(x) \approx 0$. Therefore the gradient of the squared loss is dominated by the correlation between the student and teacher model. Concretely, consider the population gradient for one parameter vector $w \in \mathbb{R}^d$ at initialization (for simplicity we omit the bias unit here):

$$\mathbb{E}_x\left[\nabla_w f_{\text{NN}}^{(0)}(x)\right] \approx -\mathbb{E}_x[x\sigma'(\langle x, w\rangle)f_*(x)]$$

$$\stackrel{(i)}{=} -\Sigma\mathbb{E}_x\left[f_*'(x)\sigma'(\langle x, w\rangle)\cdot(1+\theta)^{-1/2}\mu + f_*(x)\sigma''(\langle x, w\rangle)\cdot w\right]$$

$$\stackrel{(ii)}{=} -\sqrt{1+\theta}\mu\cdot\mathbb{E}_x[\sigma_*'(\langle x, \mu\rangle)\sigma'(\langle x, w\rangle)] + \text{residual terms},$$

where $(i)$ is due to multivariate Stein's lemma, and $(ii)$ follows from the definition of $\Sigma$. Utilizing the Hermite expansions of the student and teacher nonlinearities,

$$\sigma(z) = \sum_{i=0}^{\infty}\alpha_i h_i(z), \quad \sigma_*(z) = \sum_{i=0}^{\infty}\alpha_i^* h_i(z),$$

we have the following decomposition on the strength of the correlation term,

$$\mathbb{E}_x[\sigma_*'(\langle x, \mu\rangle)\sigma'(\langle x, w\rangle)] \approx \sum_{i=0}^{\infty}(i+1)^2\alpha_{i+1}\alpha_{i+1}^*\cdot\left\langle\sqrt{1+\theta}\mu, w\right\rangle^i.$$

This calculation illustrates that the (expected) first gradient step indeed contains the direction of the signal $\mu$. More importantly, the magnitude of the signal term is affected by two factors: $(i)$ it is amplified by a larger spike $\theta$, and $(ii)$ it vanishes when the information exponent $k$ is large, since $\langle\mu, w\rangle = \tilde{\Theta}_{\mathbb{P}}(d^{-1/2})$ at initialization. The gradient magnitude relates to the sample complexity because in order to establish learnability in the proportional regime, we need to achieve nontrivial gradient concentration using $n \asymp d$ samples. Consequently, the sufficient condition in Theorem 5 requires a larger $\theta$ when the information exponent $k$ is large, and vice versa.

**Remark.** *One caveat in the above derivation is that the Hermite coefficients of the student nonlinearity $\{\alpha_i\}_{i=1}^{\infty}$ need to be non-zero up to degree-$k$, which is not satisfied by ReLU or most commonly-used activation functions. We solve this issue by taking into account the bias units $b_i$ at initialization which "diversify" the nonlinearity, similar to [BBSS22] (see Lemma 15 for details).*

**Nonparametric learning with random biases.** After the signal direction is identified via gradient descent, we need to learn the unknown link function $\sigma_*$. We utilize random biases as a resource for univariate approximation – similar argument has appeared in many prior works [DLS22, BBSS22, BEG+22, MHPG+23, ABAM23]. Recall that in the representation learning phase, the bias units $b_i$ are not optimized. Therefore, if the weight vectors $w$ align with the signal direction $w \approx \mu$, then learning the second-layer coefficients $a$ via ridge regression can be reduced to a univariate kernel regression problem; specifically, given $x_i, x_j \in \mathbb{R}^d$, the univariate kernel is given by $k(z_i, z_j) = \mathbb{E}_{a,b}[\sigma(a\cdot z_i + b)\sigma(a\cdot z_j + b)]$, where $z_i = \langle x_i, \mu\rangle$, $z_j = \langle x_j, \mu\rangle$. For $\sigma = \text{ReLU}$, it is known that such kernel can efficiently learn a polynomial link function $\sigma_*$ (e.g., see [DLS22, BBSS22]).

# 5 Experiments: Comparing KRR and NN

In this section, we empirically validate the theoretical results presented in previous sections. We construct target functions $f_*$ with a link of varying degree $p$ and information exponent $k$ and experimentally compute the prediction risk of KRR and that of a two-layer NN.

$\boxed{k = p}$: **NN & KRR are comparable.** We first consider the setting where $k = p$, which indicates that the link function is a pure degree-$p$ Hermite polynomial: $\sigma_*(z) = h_p(z)$. Recall that $\theta \asymp d^\beta$; Theorem 3 implies that KRR learns the target function in the proportional regime when the spike magnitude satisfies $\beta > 1 - \frac{1}{p}$, which matches the sufficient condition for NN given in Theorem 5.

In the experiments of Figure 2, we compute the prediction risk of KRR and that of two-layer NN optimized via the two-stage procedure outlined in Section 2, where the link function $\sigma_*$ is a pure degree-2 and degree-3 Hermite polynomial. We observe that both KRR and NN can learn $f_*$ with linear sample complexity when the spike magnitude $\theta$ exceeds the same threshold predicted by our theory; intuitively speaking, this means that for the $k = p$ setting, NN and KRR utilize the low-dimensional structure with the same efficiency. Also, note that while Theorem 5 only provides an upper bound on the risk, the experiments demonstrate that the predicted scaling of $\theta$ is sharp.

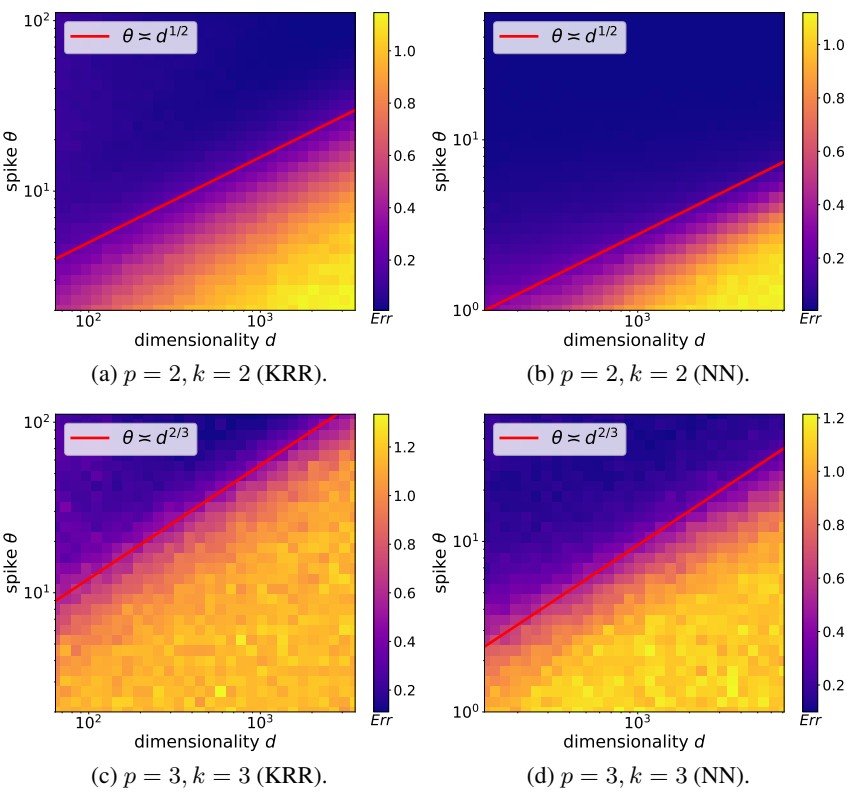

(a) $p = 2, k = 2$ (KRR).  (b) $p = 2, k = 2$ (NN).

(c) $p = 3, k = 3$ (KRR).  (d) $p = 3, k = 3$ (NN).

Figure 2: Prediction risk of KRR (a)&(c) and GD-trained two-layer NN (b)&(d) to learn $f_*$, where the link function $\sigma_*(z) = h_2(z)$ in (a)&(b), and $\sigma_*(z) = h_3(z)$ in (c)&(d). We set $\psi = 5$. For KRR we use the Gaussian RBF kernel and $\lambda = 10^{-2}$. For two-layer NN, we set $\eta = \sqrt{N}$. The experiments are averaged over 50 runs. Darker color corresponds to a smaller prediction risk, and the red lines are predicted by the sufficient conditions of learnability given by Theorem 3 and 5.

$\boxed{k < p}$: **NN outperforms KRR.** Next, we consider a setting where the link function $\sigma_*$ is a mixture of low- and high-degree components: $\sigma_*(z) = \sum_{j \in \mathcal{J}} \alpha_j h_j(z)$, with $\min \mathcal{J} = k, \max \mathcal{J} = p$. In this case, Theorem 3 predicts that KRR requires a spike magnitude of $\beta > 1 - \frac{1}{p}$ to learn $f_*$ with linear sample complexity, whereas Theorem 5 entails that a smaller spike magnitude $\beta > 1 - \frac{1}{k}$ is sufficient for the two-layer NN.

Figure 3 shows that the predictions from Theorem 3 and 5 once again accurately align with the experimental results. Observe that when $k < p$, GD-trained NN indeed outperforms KRR, in the sense that it requires a less prominent low-dimensional structure (i.e., smaller spike magnitude $\theta$) to attain linear sample complexity. This illustrates the benefit of NN under structured data due to the presence of representation (feature) learning. Roughly speaking, for target function with $k < p$, the sample complexity of representation learning (i.e., finding the signal direction $\boldsymbol{\mu}$ via gradient-based learning) is low compared to directly estimating $f_*$ on the original input features; therefore GD-trained NNs achieve better performance than KRR.

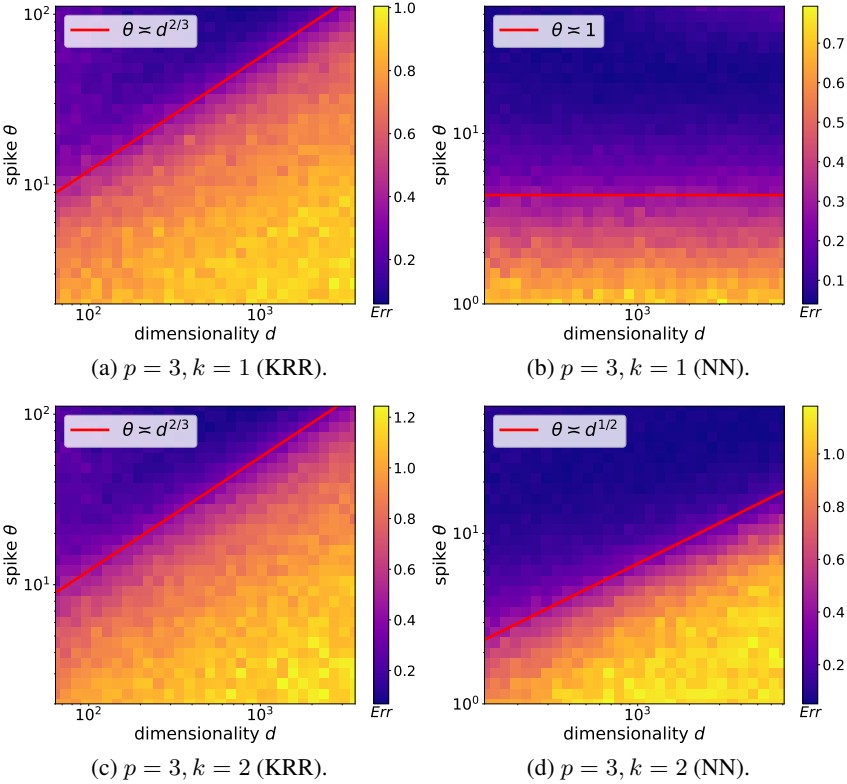

(a) $p = 3, k = 1$ (KRR).  (b) $p = 3, k = 1$ (NN).

(c) $p = 3, k = 2$ (KRR).  (d) $p = 3, k = 2$ (NN).

Figure 3: Prediction risk of KRR (a)&(c) and GD-trained two-layer NN (b)&(d) to learn $f_*$, where the link function $\sigma_*(z) = h_1(z) + h_3(z)$ in (a)&(b), and $\sigma_*(z) = h_2(z) + h_3(z)$ in (c)&(d). We use the same hyperparameters as in Figure 2. The red lines are predicted by the sufficient conditions in Theorem 3 and 5.

## 6 Conclusion and Future Directions

We investigated the performances of KRR and GD-trained NNs in the proportional asymptotic limit. We analyzed the prediction risk of these models in learning a nonlinear single-index model (2.2), where the index feature in the teacher model only relies on the spike direction $\boldsymbol{\mu}$ hidden in the anisotropic Gaussian input data (2.1). The strength $\theta$ (signal-to-noise ratio) of this spike determines the level of low dimensionality, which affects the learnability of both NNs and KRR. Our results clearly demonstrate that both NNs and KRR benefit from this additional structure in the covariance; yet, NNs adapt to the low-dimensional patterns more efficiently.

It is worth noting that our theoretical analysis focuses on the "well-specified" scenario where the input spike perfectly aligns with the index features of $f_*$. In a companion work [MHWSE23], we investigate a more general setting where the spiked covariance data only provides partial information of $f_*$ (i.e. the "misaligned" scenario), and show that feature learning (via gradient flow) also benefits from such structured data. Finally, extending our results to multi-index teacher models [CM20, DLS22, ABAM23] and considering more general training dynamics (e.g., beyond the first gradient step) applied to multiple-layer NNs are interesting directions left for future studies.

## Acknowledgement

The authors thank Enric Boix-Adsera, Joan Bruna, Bruno Loureiro, and Alireza Mousavi-Hosseini for the discussions and feedback on the manuscript. MAE was partially supported by NSERC Grant [2019-06167], CIFAR AI Chairs program, CIFAR AI Catalyst grant. TS was partially supported by JSPS KAKENHI (20H00576) and JST CREST (JPMJCR2015). ZW was partially supported by NSF DMS-2055340 and NSF DMS-2154099.

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

**Table of Contents**

# A Analysis of Kernel Ridge Regression

In this section, we analyze the performance of KRR defined in Section 2. Recall that

$$\boldsymbol{x}_i \sim \mathcal{N}(0, \boldsymbol{I}_d + \theta\boldsymbol{\mu}\boldsymbol{\mu}^\top), \quad \|\boldsymbol{\mu}\|_2 = 1, \tag{A.1}$$

where $\theta \asymp d^\beta$, and $\beta \in [0,1)$. Let $\boldsymbol{\Sigma} = \boldsymbol{I} + \theta\boldsymbol{\mu}\boldsymbol{\mu}^\top$ and $\boldsymbol{x}_i = \boldsymbol{\Sigma}^{1/2}\boldsymbol{z}_i$, where $\boldsymbol{z}_i \sim \mathcal{N}(0, \boldsymbol{I})$, and $\boldsymbol{\Sigma}^{1/2} = \boldsymbol{I} + (\sqrt{1+\theta} - 1)\boldsymbol{\mu}\boldsymbol{\mu}^\top$. Then, the training label $y_i$ can be equivalently expressed as

$$y_i = f_*(\boldsymbol{x}_i) = \sigma_*(\langle \boldsymbol{z}_i, \boldsymbol{\mu}\rangle),$$

for $i \in [n]$. Recall $\boldsymbol{X}^\top = [\boldsymbol{x}, \dots \boldsymbol{x}_n] \in \mathbb{R}^{d\times n}$. We define the nonlinear kernel matrix by $\boldsymbol{K} = g(\frac{1}{d}\boldsymbol{X}\boldsymbol{X}^\top) \in \mathbb{R}^{n\times n}$, where smooth function $g : \mathbb{R} \to \mathbb{R}$ is applied to each entry of the matrix. Additionally, we write

$$\boldsymbol{K}(\boldsymbol{x}, \boldsymbol{X}) = \left[ g(\boldsymbol{x}^\top\boldsymbol{x}_1/d), \dots, g(\boldsymbol{x}^\top\boldsymbol{x}_n/d) \right].$$

We analyze the kernel ridge regression estimator:

$$\hat{f}_{\mathrm{ker}}(\boldsymbol{x}) = \boldsymbol{K}(\boldsymbol{x}, \boldsymbol{X})(\boldsymbol{K} + \lambda\boldsymbol{I})^{-1}\boldsymbol{y}.$$

Specifically, we aim to compute the prediction risk in the proportional regime:

$$\mathcal{R}(\hat{f}_{\mathrm{ker}}) = \mathbb{E}_{\boldsymbol{x}}(\hat{f}_{\mathrm{ker}}(\boldsymbol{x}) - f_*(\boldsymbol{x}))^2,$$

where $\boldsymbol{x} \in \mathbb{R}^d$ is an independent copy of $\boldsymbol{x}_i$ sampled from (A.1). Since the kernel $\boldsymbol{K}$ is rotationally invariant, i.e.,

$$k(\boldsymbol{S}\boldsymbol{x}_i, \boldsymbol{S}\boldsymbol{x}_j) = k(\boldsymbol{x}_i, \boldsymbol{x}_j),$$

for any orthogonal matrix $\boldsymbol{S} \in \mathbb{R}^{d\times d}$ and $i, j \in [n]$, we may apply an orthogonal matrix $\boldsymbol{O} \in \mathbb{R}^{d\times d}$ for each data sample such that

$$\boldsymbol{O}\boldsymbol{x}_i \sim \mathcal{N}(0, \boldsymbol{I}_d + \theta\boldsymbol{e}_1\boldsymbol{e}_1^\top), \qquad \boldsymbol{O}\boldsymbol{x} \sim \mathcal{N}(0, \boldsymbol{I}_d + \theta\boldsymbol{e}_1\boldsymbol{e}_1^\top),$$

where $\boldsymbol{e}_1$ is the standard coordinate vector. It is clear that in distribution, $\hat{f}_{\mathrm{ker}}(\boldsymbol{x})$ and $f_*(\boldsymbol{x})$ remain unchanged before and after the transformation. Therefore, without loss of generality, we will consider the case that $\boldsymbol{\mu} = \boldsymbol{e}_1$ for the proofs in this section, that is, we write

$$\boldsymbol{\Sigma} = \boldsymbol{I}_d + \theta\boldsymbol{e}_1\boldsymbol{e}_1^\top \tag{A.2}$$

and training label $y_i = f_*(\boldsymbol{x}_i)$ only depends on the first entry of the data $\boldsymbol{x}_i$ for $i \in [n]$.

## A.1 Polynomial Approximation of Kernel Matrix

Denote the concatenation of $\boldsymbol{x}$ and $\boldsymbol{X}$ as $\tilde{\boldsymbol{X}}^\top := [\boldsymbol{x}, \boldsymbol{X}^\top] \in \mathbb{R}^{d\times(n+1)}$, and the enlarged nonlinear kernel matrix by $\tilde{\boldsymbol{K}} = g(\frac{1}{d}\tilde{\boldsymbol{X}}\tilde{\boldsymbol{X}}^\top) \in \mathbb{R}^{(n+1)\times(n+1)}$. First we present a preliminary approximation of $\tilde{\boldsymbol{K}}$ by a proper polynomial kernel.

**Lemma 6.** *Suppose that $\beta \leq 1 - \frac{1}{p}$ for some $p \in \mathbb{N}$. Under the assumptions of Theorem 3, as $n, d \to \infty$ proportionally, we have*

$$\|\tilde{\boldsymbol{H}}_p + \sigma_p\boldsymbol{I}_{n+1} - \tilde{\boldsymbol{K}}\| = o_{d,\mathbb{P}}(1),$$

*where*

$$\tilde{\boldsymbol{H}}_p := P_p(\tilde{\boldsymbol{X}}\tilde{\boldsymbol{X}}^\top/d),$$

*and $P_p(x) = \sum_{k=1}^{2p} g^{(k)}(0)x^k$, $\sigma_p := g(1) - P_p(1)$.*

**Proof.** For simplicity, we denote $\boldsymbol{x} = \boldsymbol{x}_0$ and $\boldsymbol{x}_i^\top = [\boldsymbol{x}_i^1, \bar{\boldsymbol{x}}_i^\top]$, where $\boldsymbol{x}_i^1$ is the first entry of $\boldsymbol{x}_i$, and $\bar{\boldsymbol{x}}_i$ the remaining entries for $0 \leq i \leq n$. Notice that $\frac{1}{d}\boldsymbol{x}_i^\top\boldsymbol{x}_j = \frac{1}{d}\boldsymbol{z}_i\boldsymbol{\Sigma}\boldsymbol{z}_j$, for any $0 \leq i, j \leq n$. By Bernstein inequality [Ver18, Corollary 2.8.3], we have

$$\mathbb{P}\left(\left|\frac{1}{d}\boldsymbol{x}_i^\top\boldsymbol{x}_j\right| > t\right) \leq \mathbb{P}\left(\left|\frac{1}{d}\bar{\boldsymbol{x}}_i^\top\bar{\boldsymbol{x}}_j\right| > t/2\right) + \mathbb{P}\left(\left|\frac{1}{d}\boldsymbol{x}_i^1\boldsymbol{x}_j^1\right| > t/2\right)$$

$$\leq \exp\left(-cd\min\{t^2, t\}\right) + \exp\left(-ctd^{1-\beta}\right),$$

for any $i \neq j$ and $t > 0$. Therefore, $\left|\frac{1}{d}\boldsymbol{x}_i^\top \boldsymbol{x}_j\right| = o_{d,\mathbb{P}}(d^{\beta-1}\log d)$ for $\beta \in [1/2, 1)$ and $\left|\frac{1}{d}\boldsymbol{x}_i^\top \boldsymbol{x}_j\right| = o_{d,\mathbb{P}}(d^{-1/2}\log d)$ for $\beta \in [0, 1/2]$. Then we have

$$
\begin{aligned}
\|\mathrm{offdig}(\tilde{\boldsymbol{H}}_p - \tilde{\boldsymbol{K}})\|^2 &\leq \|\mathrm{offdig}(\tilde{\boldsymbol{H}}_p - \tilde{\boldsymbol{K}})\|_F^2 \\
&\leq \sum_{i \neq j} \sum_{k=2p+1}^{\infty} \left|\frac{1}{d}\boldsymbol{x}_i^\top \boldsymbol{x}_j\right|^k = \tilde{o}_{d,\mathbb{P}}(d^{-\frac{1}{p}}).
\end{aligned}
$$

Here, $\mathrm{offdig}(\cdot)$ denotes setting the diagonal entries of the matrix to be zero. In addition, following the same proof as Theorem 1 of [EK10], we can obtain a similar approximation for the diagonal part $\mathrm{dig}(\tilde{\boldsymbol{H}}_p - \tilde{\boldsymbol{K}})$, the details of which we omit. $\qquad\square$

## A.2 Hermite Expansion of Polynomial Kernel

Equipped with the above polynomial expansion of the kernel matrix, we now introduce a further approximation using orthogonal polynomials. Recall that

$$
\mathcal{R}(\hat{f}_{\mathrm{ker}}) = \|f_*\|_{L^2}^2 - 2\boldsymbol{y}^\top (\boldsymbol{K} + \lambda \boldsymbol{I})^{-1}\boldsymbol{E} + \boldsymbol{y}^\top (\boldsymbol{K} + \lambda \boldsymbol{I})^{-1}\boldsymbol{M}(\boldsymbol{K} + \lambda \boldsymbol{I})^{-1}\boldsymbol{y},
$$

where

$$
\begin{aligned}
\boldsymbol{E} &:= \mathbb{E}_{\boldsymbol{x}}[f_*(\boldsymbol{x})k(\boldsymbol{x}, \boldsymbol{X})]^\top \in \mathbb{R}^n \\
\boldsymbol{M} &:= \mathbb{E}_{\boldsymbol{x}}[k(\boldsymbol{X}, \boldsymbol{x})k(\boldsymbol{x}, \boldsymbol{X})] \in \mathbb{R}^{n \times n}.
\end{aligned}
$$

From Lemma 6 we know that for some $p \in \mathbb{N}$,

$$
\begin{aligned}
\boldsymbol{K} &= \boldsymbol{H}_p + \boldsymbol{\Delta}_K, \\
\boldsymbol{k}(\boldsymbol{x}, \boldsymbol{X}) &= \boldsymbol{H}_p(\boldsymbol{x}, \boldsymbol{X}) + \sigma_p \boldsymbol{I}_{n+1} + \boldsymbol{\Delta}_{K'},
\end{aligned}
$$

for $\|\boldsymbol{\Delta}_K\| = o_{d,\mathbb{P}}(1)$ and $\|\boldsymbol{\Delta}_K'\| = o_{d,\mathbb{P}}(1)$, where $\sigma_p$ is defined in Lemma 6, $\boldsymbol{H}_p := P_p(\frac{1}{d}\boldsymbol{X}\boldsymbol{X}^\top)$, and $\boldsymbol{H}_p(\boldsymbol{x}, \boldsymbol{X}) := P_p(\frac{1}{d}\boldsymbol{X}\boldsymbol{x})^\top$ is a row vector. Then following from Lemmas A.3, A.8 and A.13 in [BMR21], we can conclude that $\mathcal{R}(\hat{f}_{\mathrm{ker}}) = \mathcal{R}_p + o_{d,\mathbb{P}}(1)$, where

$$
\mathcal{R}_p := \|f_*\|_{L^2}^2 - 2\boldsymbol{y}^\top (\boldsymbol{H}_p + \lambda \boldsymbol{I})^{-1}\boldsymbol{E}_p + \boldsymbol{y}^\top (\boldsymbol{K}_p + \lambda \boldsymbol{I})^{-1}\boldsymbol{M}_p(\boldsymbol{K}_p + \lambda \boldsymbol{I})^{-1}\boldsymbol{y}, \tag{A.3}
$$

and

$$
\begin{aligned}
\boldsymbol{E}_p &:= \mathbb{E}_{\boldsymbol{x}}[f_*(\boldsymbol{x})\boldsymbol{H}_p(\boldsymbol{x}, \boldsymbol{X})]^\top \in \mathbb{R}^n \\
\boldsymbol{M}_p &:= \mathbb{E}_{\boldsymbol{x}}[\boldsymbol{H}_p(\boldsymbol{X}, \boldsymbol{x})\boldsymbol{H}_p(\boldsymbol{x}, \boldsymbol{X})] \in \mathbb{R}^{n \times n}.
\end{aligned}
$$

Now we analyze the asymptotic behavior of $\mathcal{R}_p$ by utilizing the Hermitian polynomials to reorganize kernel $\boldsymbol{H}_p$. Since $\boldsymbol{\Sigma}$ now is a diagonal matrix in (A.2), from [Rah17], we know the multivariate Hermite polynomials with respect to $\mathcal{N}(0, \frac{1}{d}\boldsymbol{\Sigma})$ in $\mathbb{R}^d$ are strongly orthogonal, each of which is the product of univariate Hermite polynomials on each coordinate. More precisely, for any $k \in \mathbb{N}$ and multi-index $\boldsymbol{I} := (i_1, i_2, \ldots, i_d) \in [k]^d$ with $i_1 + i_2 + \cdots + i_d = k$, we define the $k$-th multivariate Hermite polynomial with respect to $\boldsymbol{I}$ by $h_{k,\boldsymbol{I}}(\boldsymbol{x}) := \prod_{j=1}^{d} h_{i_j}(x_j)$ where $\boldsymbol{x} = (x_1, \ldots, x_d)$ and $h_{i_j}(\cdot)$ is the $i_j$-the univariate Hermite polynomials (note that the variance for the first coordinate and the remaining coordinates are different). Hence, any $L$-degree polynomial of $\boldsymbol{x} \in \mathbb{R}^d$ has the form

$$
f(\boldsymbol{x}) = \sum_{k=0}^{L} \sum_{\substack{|\boldsymbol{I}|=k \\ \boldsymbol{I}=\{i_1, i_2, \ldots, i_d\}}} c_{k,\boldsymbol{I}} h_{k,\boldsymbol{I}}(\boldsymbol{x}), \tag{A.4}
$$

for some coefficients $c_{k,\boldsymbol{I}} \in \mathbb{R}$. Similar expansions using Gegenbauer polynomials for uniform unit sphere distribution have been used in [GMMM21, MMM21, GMMM20]. The decomposition we consider is specifically related to the Hermite expansions of $\sigma_*$:

$$
f_*(\boldsymbol{x}) = \sum_{i=k}^{p} \alpha_j h_j(\boldsymbol{\mu}^\top \boldsymbol{z}) = \sum_{i=k}^{p} \alpha_j h_j(z^1), \tag{A.5}
$$

where $h_j(\cdot)$ is the $j$-th Hermite polynomial with $z \sim \mathcal{N}(0, I)$. Notice that $\mathbb{E}[h_{k,I}(x)h_{l,J}(x)] = \delta_{k,l}\delta_{I,J}$ for $x$ sampled from the spiked model (2.1).

By the kernel trick (e.g., see Section 3.7 of [Ver18]), a $k$-th monomial kernel satisfies

$$\langle x_i, x_j \rangle^k = \langle x_i^{\otimes k}, x_j^{\otimes k} \rangle,$$

where each entry of $x_i^{\otimes k} \in \mathbb{R}^{d^k}$ is a $k$-th degree monomial of $x_i$. Hence we can rewrite the above inner product based on Hermite expansions of each entry of $x_i^{\otimes k}$ and $x_j^{\otimes k}$. Therefore, for kernel $H_p(x_i, x_j)$, there exists a Hermite polynomial feature map $\Phi : \mathbb{R}^d \to \mathbb{R}^L$ where

$$L := \sum_{k=0}^{2p} \binom{d+k-1}{k} \qquad B(d,k) := \binom{d+k-1}{k}$$

such that

$$H_p(x_i^\top x_j/d) = \Phi(x_i/\sqrt{d}) D^2 \Phi(x_i/\sqrt{d})^\top,$$

where each row vector $\Phi(\cdot)$ is the concatenation of orthogonal polynomials $h_{k,I}(\cdot)$ for $0 \leq k \leq L$ and $D$ is the diagonal matrix where the diagonal entries are the Hermite coefficients corresponding to $H_p$ which will be specified in the sequel. We define

$$\Phi := \begin{pmatrix} \Phi(x_1/\sqrt{d}) \\ \Phi(x_2/\sqrt{d}) \\ \cdots \\ \Phi(x_n/\sqrt{d}) \end{pmatrix} \in \mathbb{R}^{n \times L}, \tag{A.6}$$

which gives $H_p = \Phi\Phi^\top$. Furthermore, since the variance of the first entry of $x_i$ is different from the other entries and the label $y_i$ only depends on the first entry, we will separate the Hermite polynomials depending on $x_i^1$ and the terms depending on the remaining entries $\bar{x}_i \in \mathbb{R}^{d-1}$. That is, we further decompose each $\Phi(x_i/\sqrt{d}) = [\Phi_0(x_i/\sqrt{d}), \Phi(\bar{x}_i/\sqrt{d})]$, where $\Phi_0(x_i/\sqrt{d}) \in \mathbb{R}^{L_0}$ and

$$L_0 := \sum_{k=0}^{2p} \binom{d+k-2}{k-1} \qquad B_0(d,k) := \binom{d+k-2}{k-1},$$

where $B_0(d,k)$ is the total number of multivariate Hermite polynomials of $x_i$ with degree $k$, whose components with respect to $x_i^1$ is *not* vanishing. Based on this decomposition, we also write

$$\bar{\Phi} := \begin{pmatrix} \Phi(\bar{x}_1/\sqrt{d}) \\ \Phi(\bar{x}_2/\sqrt{d}) \\ \cdots \\ \Phi(\bar{x}_n/\sqrt{d}) \end{pmatrix}. \tag{A.7}$$

**Lemma 7.** *Let $d_{\text{eff}} := d^{1-\beta}$. Under the assumptions of Theorem 3, we know that*
$$H_p(x_i^\top x_j/d) = \Phi(x_i/\sqrt{d}) D^2 \Phi(x_i/\sqrt{d})^\top,$$
*where $D^2$ is a diagonal matrix with entries*

$$\zeta(d,k,m) = \frac{g^{(k)}(0)}{d^{k-m}d_{\text{eff}}^m} + o_d(d^{-k+m-1}d_{\text{eff}}^{-m}), \tag{A.8}$$

*for $m \in [k]$ and $k \in [2p]$. Furthermore, based on the definition of $\Phi_0$ as above, we have*

$$H_p = \Phi_0 D_0^2 \Phi_0^\top + \bar{H}_p,$$

*where each entry of $\bar{H}_p(\bar{x}_i, \bar{x}_j) := P_p(\frac{1}{d}\bar{x}_i^\top \bar{x}_j)$, and $D_0^2$ is a diagonal matrix defined analogously as $D^2$ but with $m > 0$ in (A.8). Note that $\bar{H}_p$ is an inner-product polynomial kernel matrix of $\bar{X} := [\bar{x}_1, \ldots, \bar{x}_n] \in \mathbb{R}^{(d-1) \times n}$.*

**Proof.** This lemma follows from properties of multivariate Hermite polynomials [Chi11]. Recall

$$\text{Var}(x_i^1) = 1 + \theta = \Theta(d^\beta).$$

By re-normalizing the variables, we obtain

$$P_p(x_i^\top x_j/d) = \sum_{k=0}^{2p} \sum_{m=0}^{k} \sum_{|I|=k, i_1=m} \zeta(d,k,m) h_{k,I}(x_i), h_{k,I}(x_j),$$

where $I = (i_1, \ldots, i_d)$. As for $\Phi_0$, the diagonals of $D_0$ correspond to $m \neq 0$ above. $\qquad \square$

## A.3 Analysis of Prediction Risk

The next lemma shows that $\bar{\boldsymbol{H}}_p$ is approximated by a linear kernel which does not aid the learning of $f_*$; this entails that we may focus on the term involving $\boldsymbol{\Phi}_0$. Notice that $\bar{\boldsymbol{H}}_p$ is a rotationally invariant kernel on isotropic Gaussian data $\bar{\boldsymbol{X}}$ whose limiting global law has been analyzed in [CS13].

**Lemma 8.** *Under the assumptions of Theorem 3, as $n, d \to \infty$ proportionally, we have*

$$\left\| \bar{\boldsymbol{H}}_p + \sigma_p \boldsymbol{I}_n - \boldsymbol{K}_{\mathrm{lin}} \right\| = o_{d,\mathbb{P}}(1),$$

*where*

$$\boldsymbol{K}_{\mathrm{lin}} := \frac{c_1}{d} \bar{\boldsymbol{X}}^\top \bar{\boldsymbol{X}} + c_2 \boldsymbol{I}_n + \frac{c_3}{d} \mathbf{1} \mathbf{1}^\top$$

*and $\|\boldsymbol{K}_{\mathrm{lin}}\| = \mathcal{O}_{d,\mathbb{P}}(1)$. Here, $c_1, c_2$ and $c_3$ are some positive constants that only depend on $g^{(k)}(0)$ for $k = 0, 1$ and $2$.*

The above lemma is a consequence of Theorem 1 of [EK10], and hence we omit the detailed proof; for the formulae of constants $c_1, c_2$ and $c_3$, we refer to Equation (57) of [BMR21]. We remark that in the following concentration analysis, the exact limiting spectral distribution of $\boldsymbol{K}_{\mathrm{lin}}$ does not appear, since we do not consider the exact setting of $\beta = 1 - \frac{1}{\ell}$ for $\ell \in \mathbb{N}$.

**Lemma 9.** *Under the assumptions of $f_*$ in Theorem 3, we have that*

$$\boldsymbol{y} = \boldsymbol{\Phi}_0 \boldsymbol{P} \boldsymbol{\alpha},$$

*for some vector $\boldsymbol{\alpha} = [0, \ldots, 0, \alpha_k, \ldots, \alpha_p, 0, \ldots, 0]^\top \in \mathbb{R}^{2p}$, where $\boldsymbol{P} \in \mathbb{R}^{L_0 \times 2p}$ is a projector onto the Hermite components with $\boldsymbol{I} = (i_1, 0, \ldots, 0)$.*

Here $\boldsymbol{\alpha}$ represents the Hermite coefficients of the function $f_*$. Lemma 9 directly follows from (A.5) since $\boldsymbol{y}$ only depends on $\boldsymbol{x}_i^1$ for $i \in [d]$. Notice that here $\boldsymbol{P}$ selects $h_k(\boldsymbol{x}_i^1)$ for $i \in [d]$ and $k \in [2p]$ from all multivariate Hermite polynomials.

Next, we prove the following concentration statements, parallel to existing results in [GMMM21, MMM21] for spherical data. For any $\ell \in [2p]$, we decompose $\boldsymbol{\Phi}_0 = [\boldsymbol{\Phi}_{0,\leq\ell}, \boldsymbol{\Phi}_{0,>\ell}]$ where $\boldsymbol{\Phi}_{0,\leq\ell}$ are composed of all Hermite polynomials $h_{k,\boldsymbol{I}}$ with $k \leq \ell$, and $\boldsymbol{\Phi}_{0,>\ell}$ collects all Hermite polynomials $h_{k,\boldsymbol{I}}$ with $k > \ell$. Similarly, we define $\boldsymbol{D}_{0,\leq\ell}$ and $\boldsymbol{D}_{0,>\ell}$ corresponding to $\boldsymbol{\Phi}_{0,\leq\ell} \in \mathbb{R}^{n \times m}$ and $\boldsymbol{\Phi}_{0,>\ell} \in \mathbb{R}^{n \times (L_0 - m)}$, respectively. Here we denote by $m$ the number of all Hermite polynomials $h_{k,\boldsymbol{I}}$ with $k \leq \ell$ where $m \gg n$.

**Lemma 10.** *Under the assumptions of Theorem 3, we have*

$$\boldsymbol{\Phi}_{0,\leq\ell}^\top \boldsymbol{\Phi}_{0,\leq\ell} = \boldsymbol{I}_m + o_{d,\mathbb{P}}(1).$$

This lemma shows the concentration of the Gram matrix since $\mathbb{E}[\boldsymbol{\Phi}_{0,\leq\ell}^\top \boldsymbol{\Phi}_{0,\leq\ell}] = \boldsymbol{I}_m$. The proof of this lemma can be derived from the concentration result [Zhi21]; see also Lemma 11 in [GMMM21]. The following two lemmas are the crucial ingredients of the proof of Theorem 3, which have been established in Lemmas 13 and 14 in [GMMM21] for the spherical setting.

**Lemma 11.** *Under the assumptions of Theorem 3 with $1 - \frac{1}{\ell} < \beta < 1 - \frac{1}{\ell+1}$ for some $\ell \in \mathbb{N}$, as $n, d \to \infty$ proportionally, we have*

$$\left\| n(\boldsymbol{H}_p + \lambda \boldsymbol{I}_n)^{-1} \boldsymbol{M}_p (\boldsymbol{H}_p + \lambda \boldsymbol{I}_n)^{-1} - \frac{1}{n} \boldsymbol{\Phi}_{0,\leq\ell} \boldsymbol{\Phi}_{0,\leq\ell}^\top \right\| = o_{d,\mathbb{P}}(1).$$

**Proof.** Recall the definition of $\boldsymbol{\Phi}$ in (A.6). From Lemma 7, we can obtain that $\boldsymbol{M}_p = \boldsymbol{\Phi} \boldsymbol{D}^4 \boldsymbol{\Phi}^\top$ due to the orthogonal polynomial decomposition in (A.4). Hence,

$$\boldsymbol{M} = \boldsymbol{\Phi}_{0,\leq\ell} \boldsymbol{D}_{0,\leq\ell}^4 \boldsymbol{\Phi}_{0,\leq\ell}^\top + \boldsymbol{\Phi}_{>\ell} \boldsymbol{D}_{>\ell}^4 \boldsymbol{\Phi}_{>\ell}^\top,$$

where $\boldsymbol{\Phi}_{>\ell}$ is defined similarly as $\boldsymbol{\Phi}_{0,>\ell}$ which is formed by all orthogonal components of $\boldsymbol{\Phi}_{0,\leq\ell}$. In general, $\boldsymbol{\Phi}_{>\ell}$ is composed by $\boldsymbol{\Phi}_{0,>\ell}$ and $\bar{\boldsymbol{\Phi}}$ defined in (A.7). Therefore,

$$\begin{aligned} &n(\boldsymbol{H}_p + \lambda \boldsymbol{I}_n)^{-1} \boldsymbol{M} (\boldsymbol{H}_p + \lambda \boldsymbol{I}_n)^{-1} \\ =\ & n(\boldsymbol{H}_p + \lambda \boldsymbol{I}_n)^{-1} \boldsymbol{\Phi}_{0,\leq\ell} \boldsymbol{D}_{0,\leq\ell}^4 \boldsymbol{\Phi}_{0,\leq\ell}^\top (\boldsymbol{H}_p + \lambda \boldsymbol{I}_n)^{-1} \\ &+ n(\boldsymbol{H}_p + \lambda \boldsymbol{I}_n)^{-1} \boldsymbol{\Phi}_{>\ell} \boldsymbol{D}_{>\ell}^4 \boldsymbol{\Phi}_{>\ell}^\top (\boldsymbol{H}_p + \lambda \boldsymbol{I}_n)^{-1}. \end{aligned}$$

Recall that $\boldsymbol{H}_p = \boldsymbol{\Phi}\boldsymbol{D}^2\boldsymbol{\Phi}^\top$. By Lemma 7 we know that with high probability

$$\left\| n(\boldsymbol{H}_p + \lambda\boldsymbol{I}_n)^{-1}\boldsymbol{\Phi}_{>\ell}\boldsymbol{D}_{>\ell}^4\boldsymbol{\Phi}_{>\ell}^\top(\boldsymbol{H}_p + \lambda\boldsymbol{I})^{-1} \right\| \le Cnd_{\mathrm{eff}}^{-\ell-1} = o_d(1).$$

For the second part, we let $\boldsymbol{A} := \boldsymbol{\Phi}_{>\ell}\boldsymbol{D}_{>\ell}^4\boldsymbol{\Phi}_{>\ell}^\top + \lambda\boldsymbol{I}_n$ and apply the Sherman-Morrison-Woodbury formula to obtain

$$
\begin{aligned}
&n(\boldsymbol{H}_p + \lambda\boldsymbol{I}_n)^{-1}\boldsymbol{\Phi}_{0,\le\ell}\boldsymbol{D}_{0,\le\ell}^4\boldsymbol{\Phi}_{0,\le\ell}^\top(\boldsymbol{H}_p + \lambda\boldsymbol{I}_n)^{-1} \\
&= n\boldsymbol{A}^{-1}\boldsymbol{\Phi}_{0,\le\ell}\Big(\boldsymbol{D}_{>\ell}^{-2} + \boldsymbol{\Phi}_{0,\le\ell}^\top\boldsymbol{A}^{-1}\boldsymbol{\Phi}_{0,\le\ell}\Big)^{-2}\boldsymbol{\Phi}_{0,\le\ell}^\top\boldsymbol{A}^{-1} \\
&= \boldsymbol{A}^{-1}\boldsymbol{\Phi}_{0,\le\ell}\boldsymbol{B}^2\boldsymbol{\Phi}_{0,\le\ell}^\top\boldsymbol{A}^{-1},
\end{aligned}
$$

where

$$\boldsymbol{B} := \left( (n\boldsymbol{D}_{>\ell})^{-2} + \frac{1}{n}\boldsymbol{\Phi}_{0,\le\ell}^\top\boldsymbol{A}^{-1}\boldsymbol{\Phi}_{0,\le\ell} \right)^{-1}.$$

Following the proof of Lemma 13 in [GMMM21], we can show that $\left\|\boldsymbol{A}^{-1} - \boldsymbol{I}_n\right\| = o_{d,\mathbb{P}}(1)$ and $\|\boldsymbol{B} - \boldsymbol{I}_n\| = o_{d,\mathbb{P}}(1)$ due to assumption that $1 - \frac{1}{\ell} < \ell$ and Lemma 10. $\qquad\square$

**Lemma 12.** *Recall the definition of $\boldsymbol{\Phi}_{0,\le\ell}$ above. Under the assumptions of Theorem 3 with $1 - \frac{1}{\ell} < \beta < 1 - \frac{1}{\ell+1}$ for some $\ell \in \mathbb{N}$, when $n, d \to \infty$ proportionally, we have*

$$\left\| \boldsymbol{\Phi}_{0,\le\ell}^\top(\boldsymbol{H}_p + \lambda\boldsymbol{I}_n)^{-1}\boldsymbol{\Phi}_{0,\le\ell}\boldsymbol{D}_{0,\le\ell} - \boldsymbol{I}_m \right\| = o_{d,\mathbb{P}}(1).$$

The proof of Lemma 12 is analogous to Lemma 11, the details of which we omit.

From (A.3), let us denote by

$$
\begin{aligned}
T_1 &:= \boldsymbol{y}^\top(\boldsymbol{H}_p + \lambda\boldsymbol{I})^{-1}\boldsymbol{E}_p \\
T_2 &:= \boldsymbol{y}^\top(\boldsymbol{H}_p + \lambda\boldsymbol{I})^{-1}\boldsymbol{M}_p(\boldsymbol{H}_p + \lambda\boldsymbol{I})^{-1}\boldsymbol{y}.
\end{aligned}
$$

Now we aim to control $T_1$ and $T_2$ based on Lemmas 11 and 12. We define the projection of $\boldsymbol{y}$ onto the Hermite components with degree less than or equals to $\ell$ and components with degree larger than $\ell$ as: $\boldsymbol{y} = \boldsymbol{y}_1 + \boldsymbol{y}_2$, where $\boldsymbol{y}_1 = \boldsymbol{\Phi}_{0,\le\ell}\boldsymbol{P}_{\le\ell}\boldsymbol{\alpha}$ and $\boldsymbol{y}_2 = \boldsymbol{\Phi}_{0,>\ell}\boldsymbol{P}_{>\ell}\boldsymbol{\alpha}$. Here $\boldsymbol{P}_{\le\ell}$ projects onto the Hermite components with $\boldsymbol{I} = (i_1, 0, \dots, 0)$ and degree no greater $\ell$, and $\boldsymbol{P}_{>\ell} = \boldsymbol{P} - \boldsymbol{P}_{\le\ell}$. Similarly, we decompose $\boldsymbol{E}_p$ as $\boldsymbol{E}_p = \boldsymbol{E}_{p,1} + \boldsymbol{E}_{p,2}$, where we denote

$$
\begin{aligned}
\boldsymbol{E}_{p,1} &:= \boldsymbol{\Phi}_{0,\le\ell}\boldsymbol{D}_{0,\le\ell}\boldsymbol{P}_{\le\ell}\boldsymbol{\alpha}, \\
\boldsymbol{E}_{p,2} &:= \boldsymbol{\Phi}_{0,>\ell}\boldsymbol{D}_{0,>\ell}\boldsymbol{P}_{>\ell}\boldsymbol{\alpha}.
\end{aligned}
$$

**Lemma 13.** *Under the assumptions of Theorem 3, we have*

$$T_1 = \|P_{\le\ell}f_*\|_{L^2}^2 + o_{d,\mathbb{P}}(1), \qquad T_2 = \|P_{\le\ell}f_*\|_{L^2}^2 + o_{d,\mathbb{P}}(1).$$

**Proof.** For simplicity, we first control $T_2$ and then apply some of these results to bound $T_1$. Let $T_2 = T_{21} + 2T_{22} + T_{23}$, where

$$
\begin{aligned}
T_{21} &:= \boldsymbol{y}_1^\top(\boldsymbol{H}_p + \lambda\boldsymbol{I})^{-1}\boldsymbol{M}_p(\boldsymbol{H}_p + \lambda\boldsymbol{I})^{-1}\boldsymbol{y}_1, \\
T_{22} &:= \boldsymbol{y}_1^\top(\boldsymbol{H}_p + \lambda\boldsymbol{I})^{-1}\boldsymbol{M}_p(\boldsymbol{H}_p + \lambda\boldsymbol{I})^{-1}\boldsymbol{y}_2, \\
T_{23} &:= \boldsymbol{y}_2^\top(\boldsymbol{H}_p + \lambda\boldsymbol{I})^{-1}\boldsymbol{M}_p(\boldsymbol{H}_p + \lambda\boldsymbol{I})^{-1}\boldsymbol{y}_2.
\end{aligned}
$$

Based on Lemmas 9, 11 and (A.8), we know that

$$
\begin{aligned}
T_{21} &= \frac{1}{n^2}\boldsymbol{\alpha}^\top\boldsymbol{P}_{\le\ell}^\top\boldsymbol{\Phi}_{0,\le\ell}^\top\boldsymbol{\Phi}_{0,\le\ell}\boldsymbol{\Phi}_{0,\le\ell}^\top\boldsymbol{\Phi}_{0,\le\ell}\boldsymbol{P}_{\le\ell}\boldsymbol{\alpha} \\
&= \|P_{\le\ell}f_*\|_{L^2}^2 + o_{d,\mathbb{P}}(1).
\end{aligned}
$$

For $T_{23}$, following from (23) in [GMMM21], when $d_{\mathrm{eff}}^{\ell+1} < 1$, by Markov's inequality, we have

$$
\begin{aligned}
T_{23} &= \frac{1}{n^2}\boldsymbol{\alpha}^\top\boldsymbol{P}_{>\ell}^\top\boldsymbol{\Phi}_{0,>\ell}^\top\boldsymbol{\Phi}_{0,\le\ell}\boldsymbol{\Phi}_{0,\le\ell}^\top\boldsymbol{\Phi}_{0,>\ell}\boldsymbol{P}_{>\ell}\boldsymbol{\alpha} + o_{d,\mathbb{P}}(1), \\
&= \frac{d_{\mathrm{eff}}^{\ell+1}}{n}\|P_{>\ell}f_*\|_{L^2}^2 + o_{d,\mathbb{P}}(1) = o_{d,\mathbb{P}}(1), \tag{A.9}
\end{aligned}
$$

since polynomials in $\mathbf{\Phi}_{0,\leq\ell}$ are orthogonal to polynomials in $\mathbf{\Phi}_{0,>\ell}$. Notice that

$$|T_{22}| \leq 2T_{21}^{1/2}T_{23}^{1/2} = o_{d,\mathbb{P}}(1).$$

This finishes the proof for $T_2$.

Now let us consider $T_1 = T_{11} + T_{12} + T_{13}$, where

$$T_{11} := \boldsymbol{y}_1^\top (\boldsymbol{H}_p + \lambda\boldsymbol{I})^{-1}\boldsymbol{E}_{1,p},$$
$$T_{12} := \boldsymbol{y}_2^\top (\boldsymbol{H}_p + \lambda\boldsymbol{I})^{-1}\boldsymbol{E}_{1,p},$$
$$T_{13} := \boldsymbol{y}^\top (\boldsymbol{H}_p + \lambda\boldsymbol{I})^{-1}\boldsymbol{E}_{2,p}.$$

From Lemmas 9 and 12, we directly have that

$$\begin{aligned} T_{11} &= \boldsymbol{\alpha}^\top \boldsymbol{P}_{\leq\ell}\boldsymbol{\alpha} + o_{d,\mathbb{P}}(1) \\ &= \|P_{\leq\ell}f_*\|_{L^2}^2 + o_{d,\mathbb{P}}(1). \end{aligned}$$

Applying (A.9) from above, we have $T_{12} \leq T_{23}\|P_{\leq\ell}f_*\|_{L^2}^2 = o_{d,\mathbb{P}}(1)$. Lastly, we know that

$$\begin{aligned} |T_{13}| &\leq \|\boldsymbol{y}\|\big\|(\boldsymbol{K} + \lambda\boldsymbol{I})^{-1}\big\|\|\boldsymbol{E}_2\| \\ &\leq Cn\|f_*\|_{L^2}^2 \frac{1}{d_{\text{eff}}^{\ell+1}} = o_{d,\mathbb{P}}(1), \end{aligned}$$

based on the effective dimension $d_{\text{eff}} = d^{1-\beta}$. This completes the proof of this lemma.

$\square$

Combining all the above lemmas together, Theorem 3 can be established by considering (A.3) and the following approximation

$$\begin{aligned} \mathcal{R}(f_{\text{ker}}) &= \|f_*\|_{L^2} - 2T_1 + T_2 = \|P_{>\ell}f_*\|_{L^2}^2 + o_{d,\mathbb{P}}(1) \\ &= \|P_{>\ell}f_*\|_{L^2}^2 + o_{d,\mathbb{P}}(1), \end{aligned}$$

where $P_{>\ell}$ is defined by in Section 3.

## B  Analysis of Neural Network

For the two-layer neural network, we perform the gradient-based optimization procedure outlined in Algorithm 1. This algorithm is almost identical to the two-stage representation learning procedure analyzed in recent works [BES$^+$22, DLS22, BEG$^+$22], with the only difference being the additional normalization step (B.2) to handle the anisotropy of the input data.

### B.1  Hermite Expansion of Population Gradient

We first consider one parameter vector at random initialization $\boldsymbol{w}_i \overset{\text{i.i.d.}}{\sim} \mathcal{N}(0, d^{-1}\boldsymbol{I}_d)$. In the following, we restrict ourselves to parameters in a subset defined as

$$\mathcal{E}_\tau = \left\{ i \in [N] \ \Big| \ \frac{1}{\tau\sqrt{d}} \leq |\langle\boldsymbol{\mu}, \boldsymbol{w}_i\rangle| \leq \frac{\tau}{\sqrt{d}} \right\},$$

where the positive scalar $\tau = \mathcal{O}(\text{polylog}(d))$ will be appropriately selected in the sequel. Note that due to the Gaussian initialization, setting $\tau = \Theta(1)$ ensures that a *constant* fraction of neurons $\boldsymbol{w}_i$ falls into the set $\mathcal{E}_\tau$. The gradient of $\boldsymbol{w}_i$ with respect to the population squared loss can be written as

$$\nabla_{\boldsymbol{w}_i}\Big[\mathbb{E}_{\boldsymbol{x}}\big(f_*(\boldsymbol{x}) - f_{\text{NN}}^{(0)}(\boldsymbol{x})\big)^2\Big] = \mathbb{E}_{\boldsymbol{x}}\left[\frac{1}{\sqrt{N}}a_i\big(f_{\text{NN}}^{(0)}(\boldsymbol{x}) - f_*(\boldsymbol{x})\big)\sigma'(\langle\boldsymbol{x}, \boldsymbol{w}_i\rangle + b_i)\boldsymbol{x}\right].$$

Due to our specific choice of parameter initialization (4.1), we know that the output of the student model at initialization $f_{\text{NN}}^{(0)}$ can be ignored (see Lemma 18 for justification). We, therefore, focus on

**Algorithm 1:** Gradient-based training for two-layer ReLU neural network.

**Hyperparameters:** learning rate $\eta$, $\ell_2$ regularization strength $\lambda$

**Normalized gradient descent on 1st layer:** Given i.i.d. training examples $\{(\boldsymbol{x}_j, y_j)\}_{j=1}^n$.

$$\boldsymbol{w}_i^{(1)} \leftarrow \boldsymbol{w}_i^{(0)} - \eta\sqrt{N} \cdot \nabla_{\boldsymbol{w}_i^{(0)}} \left( \frac{1}{n} \sum_{j=1}^n \left( y_j - f_{\text{NN}}^{(0)}(\boldsymbol{x}_j) \right)^2 \right) \qquad \text{(B.1)}$$

$$\boldsymbol{w}_i^{(1)} \leftarrow \frac{1}{\sqrt{\frac{1}{n} \sum_{j=1}^n \langle \boldsymbol{x}_j, \boldsymbol{w}_i^{(1)} \rangle^2}} \cdot \boldsymbol{w}_i^{(1)} \qquad \text{(B.2)}$$

**Ridge regression for 2nd layer:** Given i.i.d. training examples $\{(\tilde{\boldsymbol{x}}_j, \tilde{y}_j)\}_{j=1}^n$.

$$\hat{\boldsymbol{a}} \leftarrow \text{argmin}_{\boldsymbol{a}} \left\{ \frac{1}{n} \sum_{j=1}^n \left( \tilde{y}_j - \langle \boldsymbol{\phi}_j, \boldsymbol{a} \rangle \right)^2 + \lambda \|\boldsymbol{a}\|^2 \right\},$$

where $[\boldsymbol{\phi}_j]_i := \frac{1}{\sqrt{N}} \sigma(\langle \tilde{\boldsymbol{x}}_j, \boldsymbol{w}_i^{(1)} \rangle + \tilde{b}_i)$.

**return** prediction function $f_{\text{NN}}(\boldsymbol{x}) = \frac{1}{\sqrt{N}} \sum_{i=1}^N \hat{a}_i \sigma(\langle \boldsymbol{x}, \boldsymbol{w}_i^{(1)} \rangle + b_i)$

the following term capturing the correlation between the $i$-th student neuron and the teacher model:

$$g(\boldsymbol{w}_i) := \mathbb{E}_{\boldsymbol{x}}[f_*(\boldsymbol{x})\sigma'(\langle \boldsymbol{x}, \boldsymbol{w}_i \rangle + b_i)\boldsymbol{x}] \qquad \text{(B.3)}$$

$$\overset{(i)}{=} \boldsymbol{\Sigma}\mathbb{E}_{\boldsymbol{x}} \left[ f_*'(\boldsymbol{x})\sigma'(\langle \boldsymbol{x}, \boldsymbol{w}_i \rangle + b_i) \cdot \frac{1}{\sqrt{1+\theta}}\boldsymbol{\mu} + f_*(\boldsymbol{x})\sigma''(\langle \boldsymbol{x}, \boldsymbol{w}_i \rangle + b_i) \cdot \boldsymbol{w}_i \right],$$

$$= \underbrace{\sqrt{1+\theta}\boldsymbol{\mu} \cdot \mathbb{E}[f_*'(\boldsymbol{x})\sigma'(\langle \boldsymbol{x}, \boldsymbol{w}_i \rangle + b_i)]}_{(I)} + \underbrace{(\boldsymbol{I} + \theta\boldsymbol{\mu}\boldsymbol{\mu}^\top)\boldsymbol{w}_i \cdot \mathbb{E}[f_*(\boldsymbol{x})\sigma''(\langle \boldsymbol{x}, \boldsymbol{w}_i \rangle + b_i)]}_{(II)}.$$

where $(i)$ is due to Stein's lemma. Note that $\left\| \frac{1}{\sqrt{1+\theta}}\boldsymbol{\Sigma}^{1/2}\boldsymbol{\mu} \right\| = 1$. Therefore, if $\|\boldsymbol{\Sigma}^{1/2}\boldsymbol{w}_i\| \approx 1$ (which is entailed by the defined event $\mathcal{E}_\tau$), we may decompose the first term in the last equation via the Hermite expansion as follows (e.g., see [BBSS22]):

$$(I) = \sqrt{1+\theta}\boldsymbol{\mu} \cdot \mathbb{E}[f_*'(\boldsymbol{x})\sigma'(\langle \boldsymbol{x}, \boldsymbol{w}_i \rangle + b_i)]$$

$$\overset{(i)}{\approx} \sqrt{1+\theta}\boldsymbol{\mu} \cdot \sum_{j=0}^\infty (j+1)^2 \alpha_{j+1}^{b_i} \alpha_{j+1}^* \cdot \left\langle \sqrt{1+\theta}\boldsymbol{\mu}, \boldsymbol{w}_i \right\rangle^j, \qquad \text{(B.4)}$$

where $\alpha^{b_i}, \alpha^*$ are the Hermite coefficients of the (shifted) student and teacher nonlinearities:

$$\sigma(z + b_i) = \sum_{j=0}^\infty \alpha_j^{b_i} h_j(z), \quad \sigma_*(z) = \sum_{j=0}^\infty \alpha_j^* h_j(z).$$

The following lemma controls the error in $(i)$ due to the norm fluctuation of $\boldsymbol{w}_i$.

**Lemma 14.** *Define $\tilde{\boldsymbol{w}}_i = \boldsymbol{w}_i / \|\boldsymbol{\Sigma}^{1/2}\boldsymbol{w}_i\|$ for $i \in \mathcal{E}(\tau)$, then we have*

$$\Delta := |\mathbb{E}_{\boldsymbol{x}}[f_*'(\boldsymbol{x})\sigma'(\langle \boldsymbol{x}, \boldsymbol{w}_i \rangle + b_i)] - \mathbb{E}_{\boldsymbol{x}}[f_*'(\boldsymbol{x})\sigma'(\langle \boldsymbol{x}, \tilde{\boldsymbol{w}}_i \rangle + b_i)]| \lesssim \frac{1}{\sqrt{d}} + \frac{\tau}{d^{1-\beta}}.$$

**Proof.** By Cauchy-Schwarz inequality, we have

$$|\mathbb{E}[f_*'(\boldsymbol{x})\sigma'(\langle \boldsymbol{x}, \boldsymbol{w}_i \rangle + b_i)] - \mathbb{E}[f_*'(\boldsymbol{x})\sigma'(\langle \boldsymbol{x}, \tilde{\boldsymbol{w}}_i \rangle + b_i)]|$$

$$\leq \sqrt{\|f_*'(\boldsymbol{x})\|_{L^2}^2 \cdot \|\sigma'(\langle \boldsymbol{x}, \boldsymbol{w}_i \rangle + b_i) - \sigma'(\langle \boldsymbol{x}, \tilde{\boldsymbol{w}}_i \rangle + b_i)\|_{L^2}^2}.$$

Note that $\|f_*'(\boldsymbol{x})\|_{L^2} = \mathbb{E}[\sigma_*'(\xi)^2]^{1/2} = \mathcal{O}(1)$, where $\xi \sim \mathcal{N}(0, 1)$.

$$\|\sigma'(\langle \boldsymbol{x}, \boldsymbol{w}_i \rangle + b_i) - \sigma'(\langle \boldsymbol{x}, \tilde{\boldsymbol{w}}_i \rangle + b_i)\|_{L^2} \leq |\|\boldsymbol{\Sigma}^{1/2}\boldsymbol{w}_i\| - 1| \leq \frac{1}{\sqrt{d}} + \frac{\tau}{d^{1-\beta}},$$

which concludes the proof. $\qquad\square$

Similarly, for the second term, we have

$$
\begin{aligned}
(II) &= (\boldsymbol{I}_d + \theta\boldsymbol{\mu}\boldsymbol{\mu}^\top)\boldsymbol{w}_i \cdot \mathbb{E}[f_*(\boldsymbol{x})\sigma''(\langle\boldsymbol{x},\boldsymbol{w}_i\rangle)] \\
&\overset{(i)}{\approx} \theta\langle\boldsymbol{\mu},\boldsymbol{w}_i\rangle\boldsymbol{\mu} \cdot \sum_{j=0}^{\infty}(j+1)(j+2)\alpha_{j+2}\alpha_i^* \cdot \left\langle\sqrt{1+\theta}\boldsymbol{\mu},\boldsymbol{w}_i\right\rangle^j \\
&\quad + \boldsymbol{w}_i \cdot \sum_{j=0}^{\infty}(j+1)(j+2)\alpha_{j+2}\alpha_i^* \cdot \left\langle\sqrt{1+\theta}\boldsymbol{\mu},\boldsymbol{w}_i\right\rangle^j,
\end{aligned}
\tag{B.5}
$$

where $(i)$ can be justified using the same argument from Lemma 14, the details of which we omit.

**Random bias units "diversify" the gradient update.** Importantly, the shift introduced by the bias term $b_i$ ensures almost all neurons will have a non-vanishing Hermite coefficient at the given degree, as shown by the following lemma.

**Lemma 15.** *Given degree $p \in \mathbb{N}$, for all $b \in \mathbb{R}$ such that $h_{p-2}(b) \neq 0$ when $p \geq 2$, we have that the $p$-th Hermite coefficient of $f(z) = \sigma(z+b)$ is non-zero, i.e., $\left|\mathbb{E}_{z\sim\mathcal{N}(0,1)}[\sigma(z+b)h_p(z)]\right| > 0$.*

**Proof.** First note that when $p = 1$, we have

$$
\int_{\mathbb{R}} \mathrm{ReLU}(x+b)xe^{-\frac{x^2}{2}}\,dx = \frac{1}{2}\left(\mathrm{erf}\left(\frac{b}{\sqrt{2}}\right)+1\right) > 0.
$$

For $b \geq 2$, we divide the computation based on the parity of $p$. When $p$ is even, i.e., $p = 2s$, for any $b \in \mathbb{R}$, the Hermite coefficients are given by

$$
\int_{\mathbb{R}} \mathrm{ReLU}(x+b)h_{2s}(x)e^{-\frac{x^2}{2}}\,dx = e^{-\frac{b^2}{2}}h_{2s-2}(b),
$$

for all $s \in \mathbb{N}$. As for odd $p = 2s+1$, the odd-order Hermite coefficients are given by

$$
\int_{\mathbb{R}} \mathrm{ReLU}(x+b)h_{2s+1}(x)e^{-\frac{x^2}{2}}\,dx = -e^{-\frac{b^2}{2}}h_{2s-1}(b).
$$

Therefore, as long as $b$ is not the zero of the Hermite polynomial of degree $p-2$, the Hermite coefficients of $\mathrm{ReLU}(x+b)$ would be nonzero. $\qquad\square$

**Approximation of population gradient.** Note that for fixed $p \in \mathbb{N}$, the condition $h_{p-2}(b) \neq 0$ excludes less than $p$ possible values of bias $b_i$. Combining (B.4) and (B.5), we have the following characterization of the correlation term (B.3). Now given $k \in \mathbb{N}$ as the information exponent of $\sigma_*$, we restrict ourselves to $b_i$ such that the corresponding degree-$k$ Hermite coefficient of the shifted ReLU $\alpha_k^{b_i} = \mathbb{E}_{z\sim\mathcal{N}(0,1)}[\sigma(z+b_i)h_k(z)]$ is at least of order $\frac{1}{\log^C d}$ for some constant $C > 0$. In Section B.3.1 we show that such a condition is satisfied by a sufficiently large set of $b_i$.

**Proposition 16.** *Given fixed $k \in \mathbb{N}$ and $\theta \asymp d^\beta$ where $\beta \in [0,1)$, consider neurons $\boldsymbol{w}_i$ where the index $i \in \mathcal{E}_\tau$ for $\tau = \mathcal{O}\left(\log^C d\right)$ ensures that $\left|\alpha_k^{b_i}\right| = \Omega\left(\frac{1}{\log^C d}\right)$. Then we have*

$$
\|(I)\|^2 = \Theta\left(\alpha_k^{b_i}\langle\boldsymbol{\mu},\boldsymbol{w}_i\rangle^{2(k-1)} \cdot (1+\theta)^k\right) + \mathcal{O}(\Delta).
$$
$$
\|(II)\|^2 = \mathcal{O}\left(\mathrm{poly}(\tau) \cdot \left(d^{-k-1}(1+\theta)^{k+2} + d^{-k}(1+\theta)^k\right)\right) + \mathcal{O}(\Delta).
$$

*Where $\Delta \lesssim \frac{1}{d^{1/2}} + \frac{\tau}{d^{1-\beta}}$ is defined in Lemma 14.*

**Proof.** Since the link function $\sigma_*$ has information exponent $k$, by definition, we know that $\alpha_i^* = 0$ for all $i < k$. Also, since $\alpha_k^{b_i} = \Omega\left(\frac{1}{\log^C d}\right)$, under event $\mathcal{E}_i(\tau)$ with $\tau = \mathcal{O}\left(\log^C d\right)$, the expected

gradient is dominated by the term involving $\alpha_k^*$ in the summation. For $(I)$ we have

$$\left\| \sqrt{1+\theta}\boldsymbol{\mu} \cdot \sum_{j=0}^{\infty} (j+1)^2 \alpha_{j+1}^{b_i} \alpha_{j+1}^* \cdot \left\langle \sqrt{1+\theta}\boldsymbol{\mu}, \boldsymbol{w}_i \right\rangle^j \right\|$$

$$= (1+o_d(1)) \left\| \sqrt{1+\theta}\boldsymbol{\mu} \cdot \alpha_k^{b_i} \alpha_k^* \cdot \left\langle \sqrt{1+\theta}\boldsymbol{\mu}, \boldsymbol{w}_i \right\rangle^{k-1} \right\|$$

$$= \Theta\left( \alpha_k^{b_i} \langle \boldsymbol{\mu}, \boldsymbol{w}_i \rangle^{k-1} (1+\theta)^{\frac{k}{2}} \right).$$

The computation of $\|(II)\|$ follows from the exact same procedure. Finally, the proof is complete by including the norm fluctuation error $\Delta$ in Lemma 14. $\qquad\square$

Proposition 16 establishes that $(I)$ is the dominating term in the correlation $g(\boldsymbol{w}_i)$ for any $\beta > 1 - \frac{1}{k}$ as shown in the following corollary.

**Corollary 17.** *Under the same assumptions as Proposition 16, if $\beta > 1 - \frac{1}{k}$, we have*

$$\|g(\boldsymbol{w}_i) - (I)\| \lesssim d^{-\varepsilon} \|g(\boldsymbol{w}_i)\|,$$

*for some small constant $\varepsilon > 0$.*

## B.2  Finite-sample Concentration

Given $n$ training examples $\{(\boldsymbol{x}_i, y_i)\}_{i=1}^n$, denote the matrix of training data $\boldsymbol{X} \in \mathbb{R}^{n \times d}$, and the vector of labels $\boldsymbol{y} \in \mathbb{R}^n$. Similarly, we denote the initial first-layer parameters as $\boldsymbol{W}_0 \in \mathbb{R}^{d \times N}, \boldsymbol{b}_0 \in \mathbb{R}^N, \boldsymbol{a}_0 \in \mathbb{R}^n$. We first show that the initial gradient (B.1) is dominated by the correlation term (B.3).

**Lemma 18.** *When $n, d, N \to \infty$ proportionally, there exist some constants $c, C > 0$ such that*

$$\mathbb{P}\left( \left\| \frac{1}{n} \sum_{j=0}^n f_{\mathrm{NN}}^{(0)}(\boldsymbol{x}_j) \sigma'(\langle \boldsymbol{x}_j, \boldsymbol{w}_i \rangle + b_i) \boldsymbol{x}_j \right\| \geq C \frac{\sqrt{1+\theta}}{d} \right) \leq \exp\left( -c \log^2 d \right).$$

**Proof.** Using the matrix & vector notation,

$$\frac{1}{n} \sum_{j=0}^n f_{\mathrm{NN}}^{(0)}(\boldsymbol{x}_j) \sigma'(\langle \boldsymbol{x}_j, \boldsymbol{w}_i \rangle + b_i) \boldsymbol{x}_j = \frac{1}{n} \boldsymbol{X}^\top (f_{\mathrm{NN}}^{(0)}(\boldsymbol{X}) \odot \sigma'(\boldsymbol{X}\boldsymbol{w}_i + b_i \mathbf{1}_n)).$$

Since $\sigma'$ is bounded, we know that

$$\left\| \sum_{j=0}^n f_{\mathrm{NN}}^{(0)}(\boldsymbol{x}_j) \sigma'(\langle \boldsymbol{x}_j, \boldsymbol{w}_i \rangle + b_i) \boldsymbol{x}_j \right\| \lesssim \|\boldsymbol{X}\| \left\| f_{\mathrm{NN}}^{(0)}(\boldsymbol{X}) \right\| = \frac{1}{\sqrt{N}} \|\boldsymbol{X}\| \|\sigma(\boldsymbol{X}\boldsymbol{W}_0 + \boldsymbol{b}_0)\boldsymbol{a}_0\|.$$

To control the RHS, we first recall that the operator norm of $\boldsymbol{X}$ is bounded as follows,

$$\mathbb{P}\left( \|\boldsymbol{X}\| \geq C\sqrt{(1+\theta)d} \right) \leq \exp(-cd), \tag{B.6}$$

for some constants $c, C > 0$. Next, we consider the event that the norm of each training example $\boldsymbol{x}_j$ can be controlled as follows: $\mathcal{B} := \left\{ \left| \|\boldsymbol{x}_j\|/\sqrt{d} - 1 \right| \leq 1/2, \ j \in [n] \right\}$. Recall that $\boldsymbol{x}_j = \boldsymbol{\Sigma}^{1/2} \boldsymbol{z}_j$; hence $\|\boldsymbol{x}_j\|^2 \leq \|\boldsymbol{z}_j\|^2 + (1+\theta)|\langle \boldsymbol{z}_j, \boldsymbol{\mu} \rangle|^2$, which is clearly dominated by $\|\boldsymbol{z}_j\|^2$ when $\beta < 1$. Consequently, [Ver18, Theorem 3.1.1] implies that

$$\mathbb{P}(\mathcal{B}) \geq 1 - n \exp(-cd). \tag{B.7}$$

Under event $\mathcal{B}$, we know that $f_j := \langle \boldsymbol{a}_0, \sigma(\boldsymbol{W}_0^\top \boldsymbol{x}_j + \boldsymbol{b}_0) \rangle = \sum_{k=0}^N a_j \sigma(\langle \boldsymbol{x}_j, \boldsymbol{w}_k \rangle + b_k)$ is a sum of independent sub-exponential random variables, where the sub-exponential norm can be bounded as $\|f_j\|_{\psi_1} \leq \|a_j\|_{\psi_2} \|\sigma(\langle \boldsymbol{x}_j, \boldsymbol{w}_k \rangle + b_k)\|_{\psi_2} \lesssim N^{-1/2}$ conditioning on event $\mathcal{B}$. Therefore, for each $j \in [n]$, we have

$$\mathbb{P}(|f_j| \geq C \log d) \leq \exp\left( -c \log^2 d \right). \tag{B.8}$$

Combining (B.6), (B.7) and taking a union bound over all $f_j$ in (B.8) establishes the lemma. $\qquad\square$

The following proposition establishes the finite-sample concentration of the correlation term $g(\boldsymbol{w}_i)$.

**Proposition 19.** *Define the empirical correlation term*

$$g_n(\boldsymbol{w}_i) := \frac{1}{n}\sum_{j=1}^n \boldsymbol{x}_j f_*(\boldsymbol{x}_j)\sigma'(\langle \boldsymbol{x}_j, \boldsymbol{w}_i\rangle + b_i),$$

*and recall the population counterpart*

$$g(\boldsymbol{w}_i) = \mathbb{E}_{\boldsymbol{x}}[f_*(\boldsymbol{x})\sigma'(\langle \boldsymbol{x}, \boldsymbol{w}_i\rangle + b_i)\boldsymbol{x}].$$

*Then for any fixed $\boldsymbol{w}_i$ and $b_i$, and any $p$-degree polynomial $\sigma_*$ with information exponent $k$, if $\beta \in [0,1)$, we have*

$$\mathbb{P}\left(\|g_n(\boldsymbol{w}_i) - g(\boldsymbol{w}_i)\| \geq \log^{4p}(n)\sqrt{\frac{d}{n}}\right) \leq n\exp(-c\log^2 n), \tag{B.9}$$

*for some constant $c > 0$. Moreover, when $i \in \mathcal{E}_\tau$ and $\alpha_k^{b_i} = \Omega\left(\frac{1}{\log^C d}\right)$, we have*

$$\mathbb{P}\left(\|g_n(\boldsymbol{w}_i) - g(\boldsymbol{w}_i)\| \geq C\|g(\boldsymbol{w}_i)\|\cdot\mathrm{polylog}(n,d)\sqrt{\frac{d_{\mathrm{eff}}^k}{n}}\right) \leq n\exp(-c\log^2 n), \tag{B.10}$$

*for any $1 > \beta > 1 - \frac{1}{k}$, where $d_{\mathrm{eff}} := d^{1-\beta}$.*

**Proof.** Define the projection $\boldsymbol{P} := \boldsymbol{I}_d - \boldsymbol{\mu}\boldsymbol{\mu}^\top$, we have the following decomposition

$$\boldsymbol{\Sigma}^{1/2}\boldsymbol{z}_i = \boldsymbol{P}\boldsymbol{z}_i + \sqrt{1+\theta}\boldsymbol{\mu}\boldsymbol{\mu}^\top\boldsymbol{z}_i,$$

where Gaussian random vectors $\boldsymbol{P}\boldsymbol{z}_i$ and $(\boldsymbol{\mu}^\top\boldsymbol{z}_i)\boldsymbol{\mu}$ are centered and independent with each other. With this decomposition, we further decompose $g_n(\boldsymbol{w}_i)$ by

$$\begin{aligned}
g_n(\boldsymbol{w}_i) &= \frac{1}{n}\sum_{i=1}^n \boldsymbol{\Sigma}^{1/2}\boldsymbol{z}_i\sigma_*(\boldsymbol{\mu}^\top\boldsymbol{z}_i)\sigma'(\boldsymbol{w}^\top\boldsymbol{\Sigma}^{1/2}\boldsymbol{z}_i + b) \\
&= \underbrace{\left(\frac{1}{n}\sum_{i=1}^n \boldsymbol{P}\boldsymbol{z}_i\sigma_*(\boldsymbol{\mu}^\top\boldsymbol{z}_i)\sigma'(\boldsymbol{w}^\top\boldsymbol{\Sigma}^{1/2}\boldsymbol{z}_i + b)\right)}_{\boldsymbol{I}_1} \\
&\quad + \boldsymbol{\mu}\underbrace{\left(\frac{\sqrt{1+\theta}}{n}\sum_{i=1}^n \boldsymbol{\mu}^\top\boldsymbol{z}_i\sigma_*(\boldsymbol{\mu}^\top\boldsymbol{z}_i)\sigma'(\boldsymbol{w}^\top\boldsymbol{\Sigma}^{1/2}\boldsymbol{z}_i + b)\right)}_{\boldsymbol{I}_2}.
\end{aligned}$$

Therefore, for any $t > 0$ and $q \in (0,1)$, we have

$$\begin{aligned}
&\mathbb{P}(\|g_n(\boldsymbol{w}_i) - g(\boldsymbol{w}_i)\| > t) \\
&\leq \inf_{q\in(0,1)}\{\mathbb{P}(\|\boldsymbol{I}_1 - \mathbb{E}\boldsymbol{I}_1\| \geq qt) + \mathbb{P}(\|\boldsymbol{I}_2 - \mathbb{E}\boldsymbol{I}_2\| \geq (1-q)t)\}. \tag{B.11}
\end{aligned}$$

In the following, we present the concentration inequalities of $\boldsymbol{I}_1$ and $\boldsymbol{I}_2$ separately.

Now we first consider the concentration of $\boldsymbol{I}_2$. Notice that for any $s > 0$,

$$\mathbb{P}(\|\boldsymbol{I}_2 - \mathbb{E}\boldsymbol{I}_2\| \geq s) = \mathbb{P}\left(\sqrt{1+\theta}\left|\tfrac{1}{n}\sum_{i=1}^n f(\boldsymbol{z}_i) - \mathbb{E}[f(\boldsymbol{z}_i)]\right| \geq s\right),$$

where function $f : \mathbb{R}^d \to \mathbb{R}$ is defined by $f(\boldsymbol{z}) := \boldsymbol{\mu}^\top\boldsymbol{z}\sigma_*(\boldsymbol{\mu}^\top\boldsymbol{z})\sigma'(\boldsymbol{w}^\top\boldsymbol{\Sigma}^{1/2}\boldsymbol{z} + b)$. Based on [Sam23], we next estimate the Orlicz norm of $f(\boldsymbol{z}_i)$. Recall that the $\alpha$-exponential Orlicz norm of a random variable is defined by

$$\|X\|_{\psi_\alpha} = \inf_{t>0}\{t : \mathbb{E}[\exp(|X|^\alpha/t^\alpha)] \leq 2\}.$$

For any $m \in \mathbb{N}$ and $\boldsymbol{z} \sim \mathcal{N}(0, \boldsymbol{I}_d)$, since $\sigma_*$ is any polynomial with degree at most $p$, we have

$$\begin{aligned}
\|f(\boldsymbol{z})\|_{L^m} = \mathbb{E}[|f(\boldsymbol{z})|^m]^{1/m} &\leq \mathbb{E}_{\xi\sim\mathcal{N}(0,1)}[|\xi\sigma_*(\xi)|^m]^{1/m} \\
&\leq m^{\frac{p+1}{2}}\|\xi\sigma_*(\xi)\|_{L^2},
\end{aligned}$$

where the last inequality is due to Gaussian hypercontractivity [MMM21, Lemma 20]. Thus, random variable $f(z)$ has a finite Orlicz norm of order $\alpha := \frac{2}{p+1}$: there exists some universal constant $C > 0$ such that $\|f(z)\|_{\psi_\alpha} \leq C$. Then by Theorem 1.5 of [GSS21], we can obtain that

$$\mathbb{P}(\|\boldsymbol{I}_2 - \mathbb{E}\boldsymbol{I}_2\| \geq s) \leq 2\exp\left(-c\left(s\sqrt{\frac{n}{1+\theta}}\right)^{\frac{2}{p+1}}\right), \tag{B.12}$$

for any $s > 0$ and some universal constant $c > 0$.

Next, we prove the concentration of $\boldsymbol{I}_1$. Let $\boldsymbol{u}_i := \boldsymbol{P}\boldsymbol{z}_i\sigma_*(\boldsymbol{\mu}^\top\boldsymbol{z}_i)\sigma'(\boldsymbol{w}^\top\boldsymbol{\Sigma}^{1/2}\boldsymbol{z}_i + b)$ for $i \in [n]$. Consider a matrix $\boldsymbol{A} \in \mathbb{R}^{d \times n}$ where the $i$-th column of $\boldsymbol{A}$ is $\bar{\boldsymbol{u}}_i := \boldsymbol{u}_i - \mathbb{E}\boldsymbol{u}_i$ for $i \in [n]$. Then, for any $s > 0$ we have,

$$\mathbb{P}(\|\boldsymbol{I}_1 - \mathbb{E}\boldsymbol{I}_1\| \geq s) \leq \mathbb{P}\left(\frac{1}{\sqrt{n}}\|\boldsymbol{A}\| \geq s\right)$$

which implies that it is sufficient to analyze the concentration of $\|\boldsymbol{A}\|$. Notice that $\boldsymbol{A}$ is composed of independent centered columns $\bar{\boldsymbol{u}}_i$ with a population covariance $\boldsymbol{S} := \mathbb{E}[\bar{\boldsymbol{u}}_i\bar{\boldsymbol{u}}_i^\top]$. We can check that

$$\begin{aligned}
\|\boldsymbol{S}\| &= \sup_{\|\boldsymbol{v}\|=1} \mathrm{Var}(\boldsymbol{v}^\top\boldsymbol{u}_i) = \sup_{\|\boldsymbol{v}\|=1,\ \boldsymbol{v}\perp\boldsymbol{\mu}} \mathrm{Var}(\boldsymbol{v}^\top\boldsymbol{u}_i) \\
&\leq \sup_{\|\boldsymbol{v}\|=1,\ \boldsymbol{v}\perp\boldsymbol{\mu}} \mathbb{E}[(\boldsymbol{v}^\top\boldsymbol{u}_i)^2] = \sup_{\|\boldsymbol{v}\|=1,\ \boldsymbol{v}\perp\boldsymbol{\mu}} \mathbb{E}[(\boldsymbol{v}^\top\boldsymbol{P}\boldsymbol{z}_i)^2] \cdot \mathbb{E}[\sigma_*(\xi)^2] \leq C,
\end{aligned}$$

for some universal constant $C > 0$. Meanwhile,

$$\begin{aligned}
\|\bar{\boldsymbol{u}}_i\| &\leq \|\boldsymbol{P}\boldsymbol{z}_i\| \cdot |\sigma_*(\boldsymbol{\mu}^\top\boldsymbol{z}_i)| + \mathbb{E}[\|\boldsymbol{P}\boldsymbol{z}_i\|]\mathbb{E}[|\sigma_*(\boldsymbol{\mu}^\top\boldsymbol{z}_i)|] \\
&\leq \|\boldsymbol{z}_i\| \cdot |\sigma_*(\boldsymbol{\mu}^\top\boldsymbol{z}_i)| + \mathbb{E}[\|\boldsymbol{z}_i\|] \cdot \mathbb{E}[|\sigma_*(\xi)|] \\
&\leq \|\boldsymbol{z}_i\| \cdot |\sigma_*(\boldsymbol{\mu}^\top\boldsymbol{z}_i)| + C'\sqrt{d}, \tag{B.13}
\end{aligned}$$

where the inequality is due to the sub-Gaussian tail of the norm of the Gaussian random vector. Applying Theorem 3.1.1 of [Ver18], we know that $\|\boldsymbol{z}_i\| \leq c'\sqrt{d}$ almost surely, for some universal constant $c' > 0$. Since $\sigma_*$ is a polynomial with degree $p$, we know that $\|\sigma_*(\xi)\|_{\psi_\alpha}$ is uniformly bounded by some constant for $\alpha = 1/p$ and $\xi \sim \mathcal{N}(0,1)$. Hence, $|\sigma_*(\xi)| \leq \log^{2p}(n)$ almost surely. Together with (B.13), we can conclude that $\|\bar{\boldsymbol{u}}_i\| \leq C\log^{2p}(n)\sqrt{d}$ almost surely as $n \to \infty$. Therefore, by Theorem 5.44 in [Ver10], we have that with probability at least $1 - n\exp(-cs^2)$,

$$\frac{1}{\sqrt{n}}\|\boldsymbol{A}\| \leq (C + s\log^{2p}(n))\sqrt{\frac{d}{n}}.$$

Combining (B.12) and choosing $q = \frac{1}{2}$ and $t = \log^{4p}(n)\sqrt{\frac{d}{n}}$ in (B.11), one can easily obtain (B.9) for any $\beta \in [0,1)$. As a corollary of (B.9), applying Proposition 16, we can directly obtain (B.10). This completes the proof of the proposition.

$\square$

## B.3 Learning $f_*$ with Neural Network Features

**Univariate approximation.** We first adapt an argument from [DLS22] showing that a ReLU non-linearity can approximate a polynomial function in one dimension with the aid of random bias units.

**Lemma 20** (Corollary 3 in [DLS22], adapted). *Let $\sigma(z) = \mathrm{ReLU}(z)$, $s \sim \mathrm{Unif}(\{-1,1\})$ and $b \sim \mathcal{N}(0,1)$. Then given twice-differentiable $f : \mathbb{R} \to \mathbb{R}$, there exists some $\upsilon(s,b)$ such that for any $|z| \leq C$ with some constant $C > 0$,*

$$\mathbb{E}[\upsilon(s,b)\sigma(sz+b)] = f(z), \quad \text{and} \quad \sup_{s,b}|\upsilon(s,b)| \lesssim \exp(C).$$

**Proof.** Let $\upsilon(s,b) = \frac{e^{c^2/2}\sqrt{2\pi}}{1-e^{-3c^2/2}}\mathbf{1}_{b\in[c,2c]}$, for any $c > 0$. Then we have

$$\mathbb{E}[\upsilon(s,b)\sigma(sz+b)] = 1,$$

for all $|z| \leq c$. Now let $v(s,b) = \frac{s}{f(c)} \mathbf{1}_{b \in [c,2c]}$, for any $c > 0$, where $f(c) := \int_c^{2c} \frac{1}{\sqrt{2\pi}} e^{-b^2/2} db$. For similar reasons, we may verify that

$$\mathbb{E}[v(s,b)\sigma(sz+b)] = z,$$

for all $|z| \leq c$. Let $v(s,b) = 2\sqrt{2\pi} f''(-sb) e^{b^2/2} \mathbf{1}_{b \in [0,c]}$, for any $c > 0$, where $f \in C^2(\mathbb{R})$. Then we can easily check that

$$\mathbb{E}[v(s,b)\sigma(sz+b)] = f(z) + F(c) + G(c)z,$$

for all $|z| \leq c$, where

$$F(c) = c(f'(-c) + f'(c)) + f(0) - 2f(c),$$
$$G(c) = f'(0) - 2f'(c).$$

Notice that this construction of $v(s,b)$ approximates the function $f(z)$ using some constant and linear function perturbations. Now recalling the first two cases, we know that we can approximate any constant and linear functions on this interval, and hence we obtain the desired claim by combining all three cases together. $\qquad\square$

### B.3.1 Analysis of the Learned Feature Map

Recall the training procedure in Algorithm 1. The following lemma shows that after the first-layer parameters are optimized for one (normalized) gradient step, there exist some second-layer coefficients that achieve small prediction risk with high probability.

**Proposition 21.** *Given fixed $k \in \mathbb{N}$ and $\beta > 1 - \frac{1}{k}$, let $\mathbf{W}_1$ be the weight matrix optimized via one normalized gradient descent step defined in Algorithm 1, there exists some second-layer coefficients $\tilde{\mathbf{a}} \in \mathbb{R}^N$ such that $\tilde{f}(\mathbf{x}) = \frac{1}{\sqrt{N}} \sigma(\mathbf{W}_1^\top \mathbf{x} + \mathbf{b})\tilde{\mathbf{a}} = \frac{1}{\sqrt{N}} \sum_{i=1}^N \tilde{a}_i \sigma(a_i \langle \mathbf{x}, \mathbf{w}_i^{(1)} \rangle + b_i)$ achieves*

$$\|f_* - \tilde{f}\|_{L^2}^2 \lesssim 1/\log^C d, \ \text{and} \ \|\tilde{\mathbf{a}}\| \lesssim \log^C d,$$

*for some large constant $C > 0$, with probability $1 - \exp(-c\log^2 N)$ when $n, d \to \infty$ proportionally and $N = \Omega(d^\epsilon)$.*

**Proof.** We follow an argument similar to [BES+22, Appendix D] by selecting a suitable set of neurons that can approximate $f_*$. Throughout this proof, we take $\varepsilon > 0$ to be some small non-vanishing constant and $C > 0$ to be some large constant. At a high level, we select the neurons following the univariate approximation result in Lemma 20 to fit a polynomial target function. To this end, we need to show that after one gradient descent step, there exists a sufficiently large set of neurons $\mathbf{w}_i^{(1)}$ that align with the signal direction, i.e., $\left| \sqrt{1+\theta} \langle \mathbf{w}_i^{(1)}, \boldsymbol{\mu} \rangle \right| \approx 1$.

We first restrict ourselves to the "average-case" neurons around the equator $\mathbf{w}_i$ where the indices $i \in \mathcal{E}_\tau$. Due to the Gaussian initialization, setting $\tau = \log^2 d$ entails $\mathbb{P}(i \notin \mathcal{E}_\tau) \lesssim \exp(-\log^2 d)$.

Now we consider the subset of bias units that gives an "informative" gradient update, i.e., the $k$-th Hermite coefficient is not vanishing: $\alpha_k^{b_i} = \Omega\left(\frac{1}{\log^C d}\right)$ for some large constant $C > 0$. Recall that Lemma 15 implies that for bias unit satisfying $h_{p-2}(b_i) \neq 0$, the degree-$k$ Hermite coefficient of $f(z) = \sigma(z + b_i)$ is non-zero. Therefore, we define the index set

$$\mathcal{B}_\varepsilon = \left\{ i \in [N] \mid |h_{p-2}(b_i)| \geq \varepsilon, \ \text{if } p \geq 2 \right\}, \tag{B.14}$$

Notice that all the zeros of Hermite polynomials are real and distinct, see Theorem 3.3.1 in [Sze39]. Additionally, from Theorem 3.3.2 in [Sze39], the zeros of $h_{p+1}(x)$ are interlacing between the zeros of $h_p(x)$. From the recurrence relation of Hermite polynomials, namely,

$$h_{p+1}(x) = xh_p(x) - h_p'(x),$$

we know that $h_p'(x) \neq 0$ if $x$ is a root of $h_p$. Thus, applying the implicit function theorem, we can deduce that by choosing $\varepsilon \asymp \frac{1}{\log^C d}$ for some large $C > 0$, we attain $\alpha_k^{b_i} = \Omega\left(\frac{1}{\log^C d}\right)$, for any $i \in \mathcal{B}_\varepsilon$. Due to the Gaussian initialization of $b_i$, we know what the expected size of the set $\mathbb{E}|\mathcal{B}_\varepsilon| = \Omega(N^{1-\varepsilon'})$ for some small $\varepsilon' > 0$. Therefore, Hoeffding's inequality implies that for large enough $N = \Omega(d^\epsilon)$, we have $\left| \frac{|\mathcal{B}_\varepsilon|}{\mathbb{E}|\mathcal{B}_\varepsilon|} - 1 \right| \leq N^{-1/2+\varepsilon'}$ with probability $1 - \exp(-c\log^2 N)$.

**Gradient approximation.** For the subset of neurons with indices $i \in \mathcal{B}_\varepsilon$, we introduce a sequence of approximations to the gradient update given as

$$g_n(\boldsymbol{w}_i^{(0)}) = \frac{1}{n} \boldsymbol{X}^\top \big( f_{\mathrm{NN}}^{(0)}(\boldsymbol{X}) - f_*(\boldsymbol{X}) \big) \odot \sigma'(\boldsymbol{X} \boldsymbol{w}_i^{(0)} + b_i \mathbf{1}_n),$$

where we dropped the factor $\frac{a_i}{\sqrt{N}}$ for convenience. First recall that by Lemma 18, we can ignore the initial NN output as follows:

$$\left\| g_n(\boldsymbol{w}_i^{(0)}) - \frac{1}{n} \boldsymbol{X}^\top (f_*(\boldsymbol{X}) \odot \sigma'(\boldsymbol{X} \boldsymbol{w}_i^{(0)} + b_i \mathbf{1}_n)) \right\| \lesssim \frac{\sqrt{1+\theta}}{d}, \tag{B.15}$$

with probability $1 - \exp\big(-c \log^2 d\big)$ for some constant $c > 0$. Note that since $\beta < 1$, the RHS can always be upper-bounded by $d^{-\varepsilon}$ for some small $\varepsilon > 0$. Next, we take into account the concentration and truncation error in Propositions 19 and 16, respectively. Define the population gradient (again omitting the scalar $\frac{a_i}{\sqrt{N}}$),

$$g(\boldsymbol{w}_i^{(0)}) = \mathbb{E}_{\boldsymbol{x}} \Big[ \boldsymbol{x} f_*(\boldsymbol{x}) \sigma'(\langle \boldsymbol{x}, \boldsymbol{w}_i^{(0)} \rangle + b_i) \Big].$$

(B.10) in Proposition 19 implies that for any fixed $p \in \mathbb{N}$, $\beta > 1 - \frac{1}{k}$, and large enough $n, d$, with probability $1 - \exp\big(-c \log^2 d\big)$, we have

$$\left\| \frac{1}{n} \boldsymbol{X}^\top (f_*(\boldsymbol{X}) \odot \sigma'(\boldsymbol{X} \boldsymbol{w}_i^{(0)} + b_i \mathbf{1}_n)) - g(\boldsymbol{w}_i^{(0)}) \right\| \lesssim d^{-\varepsilon} \left\| g(\boldsymbol{w}_i^{(0)}) \right\|, \tag{B.16}$$

for some $\varepsilon > 0$. As for the population counterpart, Corollary 17 implies that for neurons with indices $i \in \mathcal{E}_\tau \cap \mathcal{B}_\varepsilon$, if $\beta > 1 - \frac{1}{k}$, then we have

$$\left\| g\big(\boldsymbol{w}_i^{(0)}\big) - \alpha_k^{b_i} \alpha_k^* \sqrt{1+\theta} \Big\langle \sqrt{1+\theta} \boldsymbol{\mu}, \boldsymbol{w}_i^{(0)} \Big\rangle^{k-1} \boldsymbol{\mu} \right\| \lesssim d^{-\varepsilon} \Big( 1 + \big\| g(\boldsymbol{w}_i^{(0)}) \big\| \Big), \tag{B.17}$$

where $\left\| g\big(\boldsymbol{w}_i^{(0)}\big) \right\| \asymp (1+\theta)^{\frac{k}{2}} \left| \Big\langle \boldsymbol{w}_i^{(0)}, \boldsymbol{\mu} \Big\rangle^{k-1} \right| = \tilde{\Theta}\Big( (1+\theta)^{\frac{k}{2}} d^{\frac{1-k}{2}} \Big)$ due to the condition $i \in \mathcal{E}_\tau$ for $\tau = \log^C d$. Combining (B.15), (B.16), and (B.17), we know that the gradient vector $g_n(\boldsymbol{w}_i^{(0)})$ achieves significant alignment with the target direction $\boldsymbol{\mu}$ when $\beta > 1 - \frac{1}{k}$.

**Normalization step.** Now we consider the normalization procedure (B.2). When $\beta > 1 - \frac{1}{k}$, (B.15), (B.16), and (B.17) imply that for neurons $\boldsymbol{w}_i$ in the subset $i \in \mathcal{E}_\tau \cap \mathcal{B}_\varepsilon$, we have

$$\boldsymbol{w}_i^{(1)} = \boldsymbol{w}_i^{(0)} - \eta a_i \cdot \frac{1}{n} \boldsymbol{X}^\top \big( f_{\mathrm{NN}}^{(0)}(\boldsymbol{X}) - f_*(\boldsymbol{X}) \big) \odot \sigma'(\boldsymbol{X} \boldsymbol{w}_i^{(0)} + b_i \mathbf{1}_n)$$

$$\overset{(i)}{=} \boldsymbol{w}_i^{(0)} + \eta a_i \cdot \Big( \alpha_k^{b_i} \alpha_k^* \sqrt{1+\theta} \Big\langle \sqrt{1+\theta} \boldsymbol{\mu}, \boldsymbol{w}_i^{(0)} \Big\rangle^{k-1} (\boldsymbol{\mu} + \boldsymbol{\Delta}) \Big),$$

where $\|\boldsymbol{\Delta}\| \lesssim d^{-\varepsilon}$ for some $\varepsilon > 0$, with probability $1 - \exp\big(-c \log^2 d\big)$. Now recall that we select a sufficiently large learning rate $\eta = \Omega(N^{1/2+\varepsilon})$ for some small $\varepsilon > 0$. Since $a_i \sim \mathcal{N}(0, 1/N)$,

$$\left\| \boldsymbol{w}_i^{(1)} - \kappa_i \boldsymbol{\mu} \right\| \lesssim d^{-\varepsilon} |\kappa_i|, \tag{B.18}$$

where $\kappa_i = \eta a_i \alpha_k^{b_i} \alpha_k^* \sqrt{1+\theta} \Big\langle \sqrt{1+\theta} \boldsymbol{\mu}, \boldsymbol{w}_i^{(0)} \Big\rangle^{k-1} = \tilde{\Theta}\Big( N^\varepsilon (1+\theta)^{\frac{k}{2}} d^{\frac{1-k}{2}} \Big)$ by the choice of $\tau \asymp \log^C d$ and $\alpha_k^{b_i} \asymp \frac{1}{\log^C d}$. Define the normalization factor

$$\chi_i = \Big( \frac{1}{n} \sum_{j=1}^n \langle \boldsymbol{x}_j, \boldsymbol{w}_i^{(1)} \rangle^2 \Big)^{1/2} = \frac{1}{\sqrt{n}} \big\| \boldsymbol{X} \boldsymbol{w}_i^{(1)} \big\|,$$

by (B.18) we know that

$$|\chi_i - \kappa_i \|\boldsymbol{X}\boldsymbol{\mu}\|| = \frac{1}{\sqrt{n}} \Big| \big\| \boldsymbol{X} \boldsymbol{w}_i^{(1)} \big\| - \kappa_i \|\boldsymbol{X}\boldsymbol{\mu}\| \Big|$$

$$\leq \frac{1}{\sqrt{n}} \Big\| \boldsymbol{X} \Big( \boldsymbol{w}_i^{(1)} - \kappa_i \boldsymbol{\mu} \Big) \Big\| \lesssim d^{-\varepsilon} (1+\theta)^{1/2} |\kappa_i|,$$

with high probability, where we used the proportional scaling of $n, d$ in the last inequality. Now note that $\frac{1}{\sqrt{n}}\|\boldsymbol{X}\boldsymbol{\mu}\| \to \sqrt{1+\theta}$ almost surely when $n, d \to \infty$ proportionally, we have

$$\left|\chi_i^{-1} - \left(\kappa_i\sqrt{1+\theta}\right)^{-1}\right| = \frac{|\chi_i - \kappa_i(1+\theta)^{1/2}|}{|\chi_i\kappa_i(1+\theta)^{1/2}|} \lesssim d^{-\varepsilon}(1+\theta)^{-1/2}|\kappa_i^{-1}|.$$

Combining the above calculations and recalling $N = \Omega(d^\epsilon)$, we arrive at the following characterization of the normalized weight vector:

$$\left\|\left(\tfrac{1}{n}\sum_{j=1}^n\langle\boldsymbol{x}_j, \boldsymbol{w}_i^{(1)}\rangle^2\right)^{-1/2} \cdot \boldsymbol{w}_i^{(1)} - \text{sign}(\kappa_i)(1+\theta)^{-1/2}\boldsymbol{\mu}\right\|$$

$$\leq \chi_i^{-1}\left\|\boldsymbol{w}_i^{(1)} - \kappa_i\boldsymbol{\mu}\right\| + \left|\text{sign}(\kappa_i)(1+\theta)^{-1/2} - \kappa_i\chi_i^{-1}\right| \lesssim d^{-\varepsilon}(1+\theta)^{-1/2}, \qquad \text{(B.19)}$$

with probability $1 - \exp(-c\log^2 d)$. Observe that (B.19) entails that after the normalization step:

$$\boldsymbol{w}_i^{(1)} \leftarrow \left(\tfrac{1}{n}\sum_{j=1}^n\langle\boldsymbol{x}_j, \boldsymbol{w}_i^{(1)}\rangle^2\right)^{-1/2} \cdot \boldsymbol{w}_i^{(1)},$$

the updated weight vector with index $i \in \mathcal{E}_\tau \cap \mathcal{B}_\varepsilon$ approximates the target direction in the sense that $\left\|\sqrt{1+\theta} \cdot \boldsymbol{w}_i^{(1)} - \text{sign}(\kappa_i)\boldsymbol{\mu}\right\| \lesssim d^{-\varepsilon}$ with high probability, for some small $\varepsilon > 0$.

**Approximation of link function.** Now we construct certain second-layer coefficients $\tilde{a}_i$ on a subset of neurons with indices $i \in \mathcal{I}$ such that the (sub)network approximates the link function $\sigma_*$. Roughly speaking, we wish to find some collection of neurons that utilizes the univariate approximation result in Lemma 20:

$$\tilde{f}(\boldsymbol{x}) := \sum_{i\in\mathcal{I}}\frac{\tilde{a}_i}{\sqrt{N}}\sigma(\langle\boldsymbol{x}, \boldsymbol{w}_i^{(1)}\rangle + b_i) \approx \mathbb{E}[\upsilon(s,b)\cdot\sigma(s\langle\boldsymbol{x}, (1+\theta)^{-1/2}\boldsymbol{\mu}\rangle + b)].$$

First, we consider the ideal setting where $\boldsymbol{w}_i$ is exactly in the direction of $\boldsymbol{\mu}$ up to sign flip, that is, we may write $\boldsymbol{w}_i = s_i(1+\theta)^{-1/2}\boldsymbol{\mu}$ for $s_i \sim \text{Unif}(\{+1, -1\})$. This allows us to reparameterize $\langle\boldsymbol{x}, (1+\theta)^{-1/2}\boldsymbol{\mu}\rangle =: z \in \mathbb{R}$, and choose the second-layer coefficients $\tilde{a}$ according to Lemma 20 on domain $|z| \leq \log\log d$. This implies that $|\upsilon(s,b)| = \mathcal{O}(\exp(\log\log d)) = \mathcal{O}(\log d)$. The corresponding "infinite-width" neural network is written as

$$f_\infty(z) = \mathbb{E}_{s\sim\text{Unif}(\{+1,-1\}),b\sim\mathcal{N}(0,1)}[\upsilon(s,b)\cdot\sigma(sz+b)].$$

From Lemma 20 we know that for $|z| \leq \log\log d$, we have $f_\infty(z) = \sigma_*(z)$.

Now we control the difference between the ideal model and the finite-width network of interest. First we restrict ourselves to $b \in \mathcal{B}_\varepsilon'$ defined as a "continuous" analogue of (B.14): $\mathcal{B}_\varepsilon' = \{b \mid |h_{p-2}(b)| \geq \varepsilon, \text{ if } p \geq 2\}$, with $\varepsilon \asymp \frac{1}{\log^C d}$ for large $C > 0$, and write the corresponding infinite-width model as

$$f_\infty^\varepsilon(z) = \mathbb{E}_{s,b}[\upsilon(s,b)\sigma(sz+b) \cdot \mathbf{1}_{b\in\mathcal{B}_\varepsilon'}].$$

By the Lipschitz property of $\sigma$, we have $|f_\infty(z) - f_\infty^\varepsilon(z)| \lesssim \sup|\upsilon(s,b)| \cdot \frac{1}{\log^C d} = o_d(1)$.

Next, we bound the error due to truncation at the interval $|z| \leq \log\log d$. Recall that $\langle\boldsymbol{x}, (1+\theta)^{-1/2}\boldsymbol{\mu}\rangle =: z \sim \mathcal{N}(0,1)$; hence we have the following evaluation

$$\mathbb{E}_{z\sim\mathcal{N}(0,1)}\left[\left(\sigma_*(z) - f_\infty^\varepsilon(z)\right)^2\right] \leq \mathbb{E}_z\left[\left(\sigma_*(z) - f_\infty(z)\right)^2\right] + \mathbb{E}_z\left[\left(f_\infty(z) - f_\infty^\varepsilon(z)\right)^2\right]$$

$$\lesssim \frac{1}{\log^C d} + \mathbb{E}_z\left[(\sigma_*(z) - f_\infty(z))^2\mathbf{1}_{|z|\geq\log\log d}\right]. \qquad \text{(B.20)}$$

To upper-bound the above expectation, observe that

$$\mathbb{E}_z[\sigma_*(z)^2\mathbf{1}_{|z|\geq\log\log d}] \lesssim \int_{|z|>\log\log d}\sigma_*(z)^2\exp(-z^2/2)\,\mathrm{d}z \lesssim \frac{1}{\log^C d}, \qquad \text{(B.21)}$$

for large $C > 0$, where we used the fact that $\sigma_*$ is a degree-$p$ polynomial with bounded $L^2$ norm in the last inequality. On the other hand, by Cauchy-Schwarz we have

$$|f_\infty(z)| \leq \sqrt{\mathbb{E}[\upsilon(s,b)^2] \cdot \mathbb{E}[\sigma(sz+b)^2]} \lesssim \sup|\upsilon(s,b)| \cdot \sqrt{\mathbb{E}[\sigma(sz+b)^2]}.$$

Therefore, by the Lipschitz property of $\sigma$ and the fact that $\sup|v(s, b)| \asymp \log d$,

$$\mathbb{E}_z[f_\infty(z)^2 \mathbf{1}_{|z| \geq \log\log d}] \lesssim \sqrt{\mathbb{E}[f_\infty(z)^4] \cdot \mathbb{E}[\mathbf{1}_{|z| \geq \log\log d}]} \lesssim \frac{1}{\log^C d}. \tag{B.22}$$

Combining (B.20), (B.21), and (B.22) yields $\mathbb{E}_{z \sim \mathcal{N}(0,1)}\left[\left(\sigma_*(z) - f_\infty^\varepsilon(z)\right)^2\right] \lesssim \frac{1}{\log^C d} = o_d(1)$ for some large $C > 0$. Now consider the finite-width model

$$\hat{f}(z) = \sum_{i \in \mathcal{I}} \frac{\tilde{a}_i}{\sqrt{N}} \sigma(s_i z + b_i) \mathbf{1}_{i \in \mathcal{B}_\varepsilon},$$

where $s_i \stackrel{\text{i.i.d.}}{\sim} \text{Unif}(\{+1, -1\})$ and $b_i \stackrel{\text{i.i.d.}}{\sim} \mathcal{N}(0, 1)$. Similar to above, we set the second-layer $\tilde{a}$ according to Lemma 20, that is, $\tilde{a}_i = \frac{\sqrt{N}}{|\mathcal{I}|} v(s_i, b_i)$.

To establish the concentration of $\hat{f}$ around $f_\infty^\varepsilon$, we first restrict $|z| \leq \log d$. Let $\phi(s, b) = \sigma(sz + b)$. Then $\|\nabla\phi(s, b)\|_\infty \leq 1 + \log d$. Hence, applying Lipschitz concentration inequality for $s$ and $b$, we have $\|\phi(s, b)\|_{\psi_2} \leq 1 + \log d$. Since uniformly $\|v(s, b)\|_\infty \leq \log d$, we know that the sub-exponential norm $\|v(s, b)\phi(s, b)\|_{\psi_1} \leq 2\log^2 d$. Thus, utilizing the Bernstein's inequality [Ver18, Theorem 2.8.1], we know that

$$\mathbb{P}\left(\left|\frac{1}{|\mathcal{I}|} \sum_{i \in \mathcal{I}} v(s_i, b_i)\sigma(zs_i + b_i)\mathbf{1}_{i \in \mathcal{B}_\varepsilon} - f_\infty^\varepsilon(z)\right| > t \,\Big|\, |z| \leq \log d\right)$$
$$\leq 2\exp\left(-\frac{c|\mathcal{I}|}{\log^4 d}\min\{t^2, t\}\right),$$

for any $t > 0$. By construction, we can construct a subset of neurons $\mathcal{I} \subset \mathcal{E}_\tau \cap \mathcal{B}_\varepsilon$ with $|\mathcal{I}| = \Theta(N)$, with probability at least $1 - \exp(-c\log^2 d)$. Since we choose $N = \Omega(d^\epsilon)$, we may take $t \asymp \frac{1}{\log^C d}$ for large $C$, and a union bound yields $\left|\hat{f}(z) - f_\infty^\varepsilon(z)\right| \lesssim \frac{1}{\log^C d}$ with probability $1 - \exp(-c\log^2 d)$ as $d, N \to \infty$, under the event that $|z| \leq \log d$. Then by the tail bound of the standard Gaussian random variable $z$, we have

$$\mathbb{E}_{z \sim \mathcal{N}(0,1)}\left[(\hat{f}(z) - f_\infty^\varepsilon(z))^2\right]$$
$$\lesssim \mathbb{E}_z\left[(\hat{f}(z) - f_\infty^\varepsilon(z))^2 \mathbf{1}_{|z| \leq \log d}\right] + \mathbb{E}_z\left[(\hat{f}(z)^2 + f_\infty^\varepsilon(z)^2)\mathbf{1}_{|z| > \log d}\right] \lesssim \frac{1}{\log^C d}, \tag{B.23}$$

with probability $1 - \exp(-c\log^2 d)$ when $d, N \to \infty$.

To control the deviation from the ideal setting, we consider neurons $\boldsymbol{w}_i^{(1)}$ where $i \in \mathcal{E}_\tau \cap \mathcal{B}_\varepsilon$ for $\tau \asymp \log^C d$ and $\varepsilon \asymp \frac{1}{\log^C d}$. Recall that for this subset of neurons, with at least probability $1 - \exp(-c\log^2 d)$ we have $\left\|\sqrt{1+\theta} \cdot \boldsymbol{w}_i^{(1)} - \text{sign}(\kappa_i)\boldsymbol{\mu}\right\| \lesssim d^{-\varepsilon}$, where $\text{sign}(\kappa_i) \sim \text{Unif}(\{+1, -1\})$ due to the Gaussian initialization of $a_i$ (note that $\boldsymbol{w}_i, a_i, b_i$ are independent). Therefore,

$$\mathbb{E}_{\boldsymbol{x}}\left(\sum_{i \in \mathcal{I}} \frac{1}{\sqrt{N}} \tilde{a}_i \sigma(\langle \boldsymbol{x}, \boldsymbol{w}_i^{(1)}\rangle + b_i) - \sum_{i \in \mathcal{I}} \frac{1}{\sqrt{N}} \tilde{a}_i \sigma(\text{sign}(\kappa_i)\langle \boldsymbol{x}, (1+\theta)^{-1/2}\boldsymbol{\mu}\rangle + b_i)\right)^2$$

$$\stackrel{(i)}{\leq} \frac{1}{N}\|\tilde{\boldsymbol{a}}\|^2 \cdot \mathbb{E}_{\boldsymbol{x}}\left[\sum_{i \in \mathcal{I}}\langle \boldsymbol{x}, \boldsymbol{w}_i^{(1)} - \text{sign}(\kappa_i)(1+\theta)^{-1/2}\boldsymbol{\mu}\rangle^2\right] \lesssim \frac{\|\tilde{\boldsymbol{a}}\|^2|\mathcal{I}|}{Nd^\varepsilon}, \tag{B.24}$$

for small constant $\varepsilon > 0$, where we used the Lipschitz property of $\sigma$ in $(i)$.

Finally, recall that with high probability we can construct $\mathcal{I}$ such that $|\mathcal{I}| = |\mathcal{E}_\tau \cap \mathcal{B}_\varepsilon| \asymp N$. This implies the following $\ell_2$-norm control of the chosen coefficients,

$$\sum_{i \in \mathcal{I}} \tilde{a}_i^2 \lesssim |\mathcal{I}| \cdot (|\mathcal{I}|^{-1}\log d\sqrt{N})^2 = N\log^2 d \cdot |\mathcal{I}|^{-1} \lesssim \log^C d, \tag{B.25}$$

This entails that the RHS of (B.24) can be bounded as $d^{-\varepsilon}\log^C N = o_d(1)$.

Combining (B.20), (B.23), (B.24), and (B.25), we know that there exists some second-layer coefficients $\tilde{a}$ such that as $n, d, N \to \infty$, there exists some some large constant $C > 0$ such that

$$\mathbb{E}_{\boldsymbol{x}}\left[(f_*(\boldsymbol{x}) - \tilde{f}(\boldsymbol{x}))^2\right] \lesssim \frac{1}{\log^C d}, \quad \text{where } \tilde{f}(\boldsymbol{x}) = \frac{1}{\sqrt{N}}\sum_{i=1}^N \tilde{a}_i \sigma(\langle \boldsymbol{x}, \boldsymbol{w}_i^{(1)}\rangle + b_i), \ \|\tilde{\boldsymbol{a}}\| \lesssim \log^C d,$$

with probability $1 - \exp(-c\log^2 d)$. Note here here we simply set $\tilde{a}_i = 0$ for $i \notin \mathcal{E}_\tau \cap \mathcal{B}_\varepsilon$. $\qquad\square$

### B.3.2 Ridge Regression Estimator

Proposition 21 suggests there exists some second-layer coefficients with small $\ell_2$ norm can achieve small prediction risk. Based on a standard concentration argument, we can show that the ridge regression estimator with appropriately chosen $\lambda$ also achieves low risk (e.g., see [Bac23]). Here we briefly sketch an argument following [BES$^+$22, Appendix D.3].

Given feature map $\boldsymbol{x} \to \frac{1}{\sqrt{N}}\sigma(\boldsymbol{W}_1^\top \boldsymbol{x} + \boldsymbol{b})$ conditioned on first-layer weights $\boldsymbol{W}_1, \boldsymbol{b}$, we denote the associated Hilbert space as $\mathcal{H}$. We define the optimal predictor in the RKHS as $\check{f} := \operatorname{argmin}_{f \in \mathcal{H}} \mathbb{E}_{\boldsymbol{x}}(f(\boldsymbol{x}) - f_*(\boldsymbol{x}))^2$, which takes the form of $\check{f}(\boldsymbol{x}) = \langle \frac{1}{\sqrt{N}}\sigma(\boldsymbol{W}_1^\top \boldsymbol{x} + \boldsymbol{b}), \check{\boldsymbol{a}} \rangle$ for some $\check{\boldsymbol{a}} \in \mathbb{R}^N$. We have the orthogonal decomposition of $f_*$ in $L^2$:

$$f_*(\boldsymbol{x}) = \check{f}(\boldsymbol{x}) + f_\perp(\boldsymbol{x}).$$

Given fresh training data $\{(\tilde{\boldsymbol{x}}_i, \tilde{y}_i)\}_{i=1}^n$, it is known that the prediction risk of ridge regression admits the following decomposition:

$$\mathcal{R} = \mathbb{E}_{\boldsymbol{x}}\left( \left(f_*(\boldsymbol{x}) - \check{f}(\boldsymbol{x})\right) + \left(\check{f}(\boldsymbol{x}) - \frac{1}{n}\phi_x\left(\widehat{\boldsymbol{\Sigma}}_\Phi + \lambda\boldsymbol{I}\right)^{-1}\boldsymbol{\Phi}^\top \tilde{\boldsymbol{y}}\right)\right)^2$$

$$\lesssim \underbrace{\mathbb{E}_{\boldsymbol{x}}\left(f_*(\boldsymbol{x}) - \check{f}(\boldsymbol{x})\right)^2}_{R_1} + \underbrace{\mathbb{E}_{\boldsymbol{x}}\left(\check{f}(\boldsymbol{x}) - \frac{1}{n}\phi_x\left(\widehat{\boldsymbol{\Sigma}}_\Phi + \lambda\boldsymbol{I}\right)^{-1}\boldsymbol{\Phi}^\top \check{\boldsymbol{f}}\right)^2}_{R_2}$$

$$+ \underbrace{\frac{1}{n^2}\boldsymbol{f}_\perp^\top \boldsymbol{\Phi}\left(\widehat{\boldsymbol{\Sigma}}_\Phi + \lambda\boldsymbol{I}\right)^{-1}\boldsymbol{\Sigma}_\Phi\left(\widehat{\boldsymbol{\Sigma}}_\Phi + \lambda\boldsymbol{I}\right)^{-1}\boldsymbol{\Phi}^\top \boldsymbol{f}_\perp}_{R_3},$$

where $\phi_{\boldsymbol{x}} := \frac{1}{\sqrt{N}}\sigma(\boldsymbol{W}_1^\top \boldsymbol{x} + \boldsymbol{b})$ for $\boldsymbol{x} \in \mathbb{R}^d$, which gives $\boldsymbol{\Phi}^\top = [\phi_{\tilde{\boldsymbol{x}}_1}^\top, \ldots, \phi_{\tilde{\boldsymbol{x}}_i}^\top, \ldots, \phi_{\tilde{\boldsymbol{x}}_n}^\top]$, where $\tilde{\boldsymbol{x}}_i$ is the $i$-th row of $\tilde{\boldsymbol{X}}$; we denoted $\widehat{\boldsymbol{\Sigma}}_\Phi := \frac{1}{n}\boldsymbol{\Phi}^\top\boldsymbol{\Phi}$, $\boldsymbol{\Sigma}_\Phi := \frac{1}{N}\mathbb{E}_{\boldsymbol{x}}\left[\sigma(\boldsymbol{W}_1^\top \boldsymbol{x} + \boldsymbol{b})\sigma(\boldsymbol{W}_1^\top \boldsymbol{x} + \boldsymbol{b})^\top\right]$. Also, the $i$-th entry of vector $\check{\boldsymbol{f}}$ and $\boldsymbol{f}_\perp$ are given by $[\check{\boldsymbol{f}}]_i = \check{f}(\tilde{\boldsymbol{x}}_i)$, $[\boldsymbol{f}_\perp]_i = f_\perp(\tilde{\boldsymbol{x}}_i)$, respectively.

**Concentration of feature covariance.** We first establish the concentration of the kernel matrix in the proportional regime. The following lemma provides an estimate of the norm of the features.

**Lemma 22.** *Given the weight matrix after one gradient step $\boldsymbol{W}_1$ as specified in Algorithm 1, then we have $\left\|\boldsymbol{\Sigma}^{1/2}\boldsymbol{W}_1\right\|_F \lesssim \sqrt{N}$, with probability $1 - \exp\left(-c\log^2 d\right)$, as $n, d, N \to \infty, n/d \to \psi > 1$.*

**Proof.** Write $\boldsymbol{v}_i = \boldsymbol{\Sigma}^{1/2}(\boldsymbol{w}_i^{(0)} - \eta a_i \cdot g_n(\boldsymbol{w}_i^{(0)}))$, by definition $\boldsymbol{x} = \boldsymbol{\Sigma}^{1/2}\boldsymbol{z}$ for $\boldsymbol{z} \sim \mathcal{N}(0, \boldsymbol{I}_d)$, and

$$\left\|\boldsymbol{\Sigma}^{1/2}\boldsymbol{w}_i^{(1)}\right\|^{-2} = \frac{\frac{1}{n}\boldsymbol{v}^\top \boldsymbol{Z}^\top \boldsymbol{Z}\boldsymbol{v}}{\|\boldsymbol{v}\|^2} \geq \lambda_{\min}\left(\frac{1}{n}\boldsymbol{Z}^\top \boldsymbol{Z}\right),$$

which implies that $\left\|\boldsymbol{\Sigma}^{1/2}\boldsymbol{w}_i^{(1)}\right\|^2 \leq \lambda_{\min}^{-1}\left(\frac{1}{n}\boldsymbol{Z}^\top \boldsymbol{Z}\right) \lesssim 1$ almost surely as $n, d \to \infty$, $n/d > 1$. Summing over $\boldsymbol{w}_i^{(1)}$ yields the desired claim. $\qquad\square$

Observe that the feature vector $\phi_{\boldsymbol{x}} = \frac{1}{\sqrt{N}}\sigma(\boldsymbol{W}_1^\top \boldsymbol{x} + \boldsymbol{b}) = \frac{1}{\sqrt{N}}\sigma(\boldsymbol{W}_1^\top \boldsymbol{\Sigma}^{1/2}\boldsymbol{z} + \boldsymbol{b})$ is a sub-Gaussian vector with parameter $\frac{1}{\sqrt{N}}\left\|\boldsymbol{\Sigma}^{1/2}\boldsymbol{W}_1\right\|_F \lesssim 1$ due to the Lipschitzity of $\sigma$; hence we may apply the following sub-Gaussian concentration result.

**Lemma 23.** *Under the same assumption as Lemma 22, as $d \to \infty$, we have $\|\phi_{\boldsymbol{x}}\| \leq \log d$ almost surely.*

**Proof.** Notice that $\|\cdot\|$ and $\sigma(\cdot)$ are 1-Lipschitz convex functions. Following [WZ21, Section 3], we can utilize the convex concentration inequality (see [Ver18, Chapter 5]) to obtain

$$\mathbb{P}(|\|\phi_{\boldsymbol{x}}\| - \mathbb{E}_{\boldsymbol{x}}[\|\phi_{\boldsymbol{x}}\|]| > t) \leq 2\exp\left(-ct^2\right), \tag{B.26}$$

for any $t > 0$, conditioning on the event that $\left\|\boldsymbol{\Sigma}^{1/2}\boldsymbol{W}_1\right\| \leq \left\|\boldsymbol{\Sigma}^{1/2}\boldsymbol{W}_1\right\|_F \lesssim \sqrt{N}$. From Lemma 22, we know this event that $\left\|\boldsymbol{\Sigma}^{1/2}\boldsymbol{W}_1\right\|_F \lesssim \sqrt{N}$ occurs with probability at least $1 - \exp\left(-c\log^2 d\right)$. Furthermore, under this event, we have

$$\mathbb{E}_{\boldsymbol{x}}[\|\boldsymbol{\phi}_{\boldsymbol{x}}\|]^2 \leq \mathbb{E}[\|\boldsymbol{\phi}_{\boldsymbol{x}}\|^2] = \frac{1}{N}\left\|\boldsymbol{\Sigma}^{1/2}\boldsymbol{W}_1\right\|_F^2 \mathbb{E}[\sigma(\xi)^2] \lesssim 1.$$

Thus, we can take $t = \log d$ and combine Lemma 22 and (B.26) to attain the assertion. $\qquad\square$

**Lemma 24** ([Min17], adapted). *Under the same assumption as Lemma 22, given $n\lambda \geq c\log^2 d$ for some constant $c > 0$, we have*

$$\mathbb{P}\left(\left\|(\boldsymbol{\Sigma}_\Phi + \lambda\boldsymbol{I})^{-1/2}(\boldsymbol{\Sigma}_\Phi - \widehat{\boldsymbol{\Sigma}}_\Phi)(\boldsymbol{\Sigma}_\Phi + \lambda\boldsymbol{I})^{-1/2}\right\| \geq t\right) \leq Cn\exp\left(-\frac{n\lambda t^2}{2(1+t/3)\log^2 d}\right),$$

*if $t^2 \geq \log^2 d \cdot (1 + t/3)/\lambda n$.*

**Proof.** Let us denote

$$\boldsymbol{G}_i = \frac{1}{n}(\boldsymbol{\Sigma}_\Phi + \lambda\boldsymbol{I})^{-1/2}(\boldsymbol{\phi}_{\boldsymbol{x}_i}\boldsymbol{\phi}_{\boldsymbol{x}_i}^\top - \boldsymbol{\Sigma}_\Phi)(\boldsymbol{\Sigma}_\Phi + \lambda\boldsymbol{I})^{-1/2}.$$

Notice that $\mathbb{E}[\boldsymbol{G}_i] = 0$. From Lemma 23, we know that $\|[\boldsymbol{G}_i\| \leq \log^2 d/\lambda n$ almost surely. Moreover, we may verify that $\mathbb{E}[\boldsymbol{G}_i^2] \preccurlyeq \frac{1}{n}\frac{\log^2 d}{\lambda n}\boldsymbol{I}_N$ and

$$\operatorname{Tr}\mathbb{E}[\boldsymbol{G}_i^2] \leq \frac{1}{n}\frac{\log^2 d}{\lambda n}\operatorname{Tr}(\boldsymbol{\Sigma}_\Phi + \lambda\boldsymbol{I})^{-1}\boldsymbol{\Sigma}_\Phi \leq \frac{\log^2 d}{\lambda n}.$$

Here we use the fact that all the eigenvalues of $(\boldsymbol{\Sigma}_\Phi + \lambda\boldsymbol{I})^{-1}\boldsymbol{\Sigma}_\Phi$ are between 0 and 1. Thus, we can apply Matrix Bernstein bound (see (3.9) in [Min17]) to conclude this concentration result. $\qquad\square$

With Lemma 24 we have the following useful corollary.

**Corollary 25.** *Given $n\lambda \gg \log^4 d$, for any $t \in (0, 1)$, for sufficiently large $d$,*

$$\left\|(\boldsymbol{\Sigma}_\Phi + \lambda\boldsymbol{I})^{-1/2}(\boldsymbol{\Sigma}_\Phi - \widehat{\boldsymbol{\Sigma}}_\Phi)(\boldsymbol{\Sigma}_\Phi + \lambda\boldsymbol{I})^{-1/2}\right\| = o_d(1),$$

*with probability at least $1 - n\exp\left(-c\log^2 d\right)$.*

**Prediction risk of ridge regression.** We upper-bound the prediction risk of the ridge regression estimator by controlling the decomposed error terms separately.

**Proposition 26.** *Under the training procedure described in Algorithm 1 with $N = \Omega(d^\varepsilon)$ and $d^{\varepsilon-1} \lesssim \lambda \lesssim d^\varepsilon$ for small $\varepsilon > 0$, if $\beta > 1 - \frac{1}{k}$, then we have $\mathcal{R}(\hat{f}_{\mathrm{NN}}) = o_d(1)$ with probability 1 when $n, d \to \infty, n/d \to \psi$.*

**Proof.** First note that $R_1 = \mathbb{E}_{\boldsymbol{x}}\left(f_*(\boldsymbol{x}) - \check{f}(\boldsymbol{x})\right)^2 = \|f_\perp\|_{L^2}^2$ by definition. For $R_2$, [BES$^+$22, Appendix D.3] showed that $R_2 \lesssim \|f_* - \tilde{f}\|_{L^2}^2 + \lambda\|\tilde{f}\|_{\mathcal{H}}^2$. Finally, for $R_3$ due to the concentration in Corollary 25, we have the following evaluation,

$$\begin{aligned}
R_3 &= \frac{1}{n^2}\boldsymbol{f}_\perp^\top\boldsymbol{\Phi}\left(\widehat{\boldsymbol{\Sigma}}_\Phi + \lambda\boldsymbol{I}\right)^{-1}\boldsymbol{\Sigma}_\Phi\left(\widehat{\boldsymbol{\Sigma}}_\Phi + \lambda\boldsymbol{I}\right)^{-1}\boldsymbol{\Phi}^\top\boldsymbol{f}_\perp \\
&\leq \frac{1}{n}\left\|\frac{1}{\sqrt{n}}\boldsymbol{f}_\perp^\top\boldsymbol{\Phi}\left(\widehat{\boldsymbol{\Sigma}}_\Phi + \lambda\boldsymbol{I}\right)^{-1/2}\right\|^2 \cdot \left\|\left(\widehat{\boldsymbol{\Sigma}}_\Phi + \lambda\boldsymbol{I}\right)^{-1/2}(\boldsymbol{\Sigma}_\Phi - \widehat{\boldsymbol{\Sigma}}_\Phi)\left(\widehat{\boldsymbol{\Sigma}}_\Phi + \lambda\boldsymbol{I}\right)^{-1/2}\right\| \\
&\quad + \frac{1}{n}\left\|\frac{1}{n}\boldsymbol{f}_\perp^\top\boldsymbol{\Phi}\left(\widehat{\boldsymbol{\Sigma}}_\Phi + \lambda\boldsymbol{I}\right)^{-1}\boldsymbol{\Phi}^\top\right\|^2 \leq \frac{1}{n}\|\boldsymbol{f}_\perp\|^2,
\end{aligned}$$

Recall that in Proposition 21 we constructed some $\tilde{f}$ that achieves $\|f_* - \tilde{f}\|_{L^2}^2 \lesssim 1/\log^C d$ and $\|\tilde{f}\|_{\mathcal{H}}^2 \lesssim \log^C d$. This implies that $\|f_\perp\|_{L^2}^2 \lesssim 1/\log^C d$ and hence $R_1 = o_{d,\mathbb{P}}(1)$. Similarly, for $\lambda = \mathcal{O}(d^{-\varepsilon})$, we have $R_2 = o_{d,\mathbb{P}}(1)$. The proof is completed by Markov's inequality on $R_3$. $\qquad\square$

