# OpenReview forum: "Learning in the Presence of Low-dimensional Structure: A Spiked Random Matrix Perspective"
_NeurIPS.cc/2023/Conference — NeurIPS 2023 poster_

### Official Review · Reviewer_qX6D · 2023-06-21

**Soundness:** 3 good
**Presentation:** 3 good
**Contribution:** 2 fair
**Rating:** 6
**Confidence:** 3

**Summary:**

This manuscript presents an exploration of high-dimensional regression using a theoretical approach. It focuses on a model characterized by:
- A single-index model where the output is dependent on a one-dimensional projection of the input in a specific direction, $\mu$.
- Spiked covariance data with Gaussian input that exhibits variance significantly larger in the $\mu$ direction.

The authors build upon this theory in an asymptotic framework where the dimension $d$, the number of samples $n$, and the variance of the spike approach infinity. They examine the performance of kernel methods and one hidden-layer neural networks in this context, particularly when n and d are proportional. The networks are simplified for the sake of analysis, having been trained with only one gradient step in the inner layer.

The key theoretical findings suggest that neural networks efficiently learn high-degree components, provided there are low-degree components in the same direction. Conversely, it is more challenging for kernel methods to learn high-degree components under similar conditions. This observation underscores an advantage of neural networks over kernel methods. These results are backed by numerical simulations in the same idealized setting, indicating a strong agreement with the theory.

**Strengths:**

This manuscript is well-articulated and provides transparent results. To the best of my understanding, it is the first work to explore this specific model, providing insights into the limitations of neural networks and kernel methods.

**Weaknesses:**

While this study builds upon previous works that examined single-index models and/or spiked covariance data, I encourage the authors to delve deeper into the following areas:

(1) Why is the considered model better than those considered before? Could the authors identify a dataset that is likely to present a single-index model and spiked covariance data in the same direction, even approximately? The [GMMM20] work furnished an example to support their research, however, their example seemed to advocate for a multi-index model. I understand that your study deviates from theirs in the focus on the single-index scenario.

(2) While the results are appropriately compared with related findings, there is a shortage of comparison in terms of the methods of analysis. Can you highlight the novelty of your analyses compared to previous works, particularly in contrast to [GMMM20], which seems closely related to your study? How harder was it to tackle the single-index case rather than multi-index with a diverging number of directions?

Furthermore, I found it challenging to navigate through the appendix of this paper due to a plethora of typos and inaccuracies. Several sections were validated by aligning with another paper's proof (e.g., l.492-493, l.550, l.568-569, l.584-585). This raises questions about the originality of the proof techniques. As a consequence of all of this, I cannot confirm the accuracy of the proofs.

I provide detailed comments and potential corrections below to enhance the clarity and correctness of the work.

**Questions:**

Comments:
- l.104-105: This sentence is unclear. Could you elaborate?
- Eq. (2.3): It seems the notations $X$ or $k(x,X)$ have not been introduced at this stage.
- l.165: Could you clarify the "i.e." part?
- In Section 5: How would the neural network training plots appear if you used plain GD/SGD? Would the predicted scalings remain valid?
- l.475-476: This sentence seems to lack clarity and precision. A better wording could be, "The distribution of the random variable $(f_{ker}(x), f_*(x))$ remains unaltered through this rotation."

Potential Mathematical Corrections:
- Display of Theorem 3: It appears there is a square missing on the norm. Also, could we prove the stronger statement $\Vert f_{ker} - P_{>l} f_* \Vert \to 0$?
- Second equality of l.255: Considering $x$ is multidimensional, shouldn't $f'_*(x)$ be a vector?
- Display of l.258: Shouldn't there be (i+1) factors in the sum?
- l.483: p=0 seems to be an invalid choice as 1/p appears.
- Display of l.489: Based on Prop. 2.7.1(a) of [Ver18], shouldn't there be 2 factors?
- Display of l.491: Should there be constants in the inequality?
- Second equation of l.499: The dimensions do not seem to match.
- l.506: Considering the variance in one direction is not equal to 1, shouldn't the formula for the Hermite polynomials be different?

Typographical Errors:
- l.100: The correct term should be "teacher" instead of "teaching".
- l.206: Please replace "expend" with "expand".
- l.207: "bases" should be replaced with "basis".
- l.467: Considering the regression setting, it would be more accurate to use "output" instead of "label".
- l.480-481: This does not form a complete sentence.
- l.484: Remove one "+" sign.
- Second equation of l.499: It should be \Delta_K' (not \Delta_{K'}).
- l.501: Shouldn't it be "Hermite" instead of "Hermitian"?


**Limitations:**

Yes

---

> ### Author Rebuttal · Authors · 2023-08-09
>
> Thank you for the detailed and thoughtful feedback. We address the technical points below.
>
> **Q: Motivation of single-index $f_*$**
>
> The single-index model is a canonical setting of learning low-dimensional signals from high-dimensional data; it covers many statistical learning problems such as phase retrieval and generalized linear models. Importantly, the low dimensionality of the single-index target may be exploited by *adaptive* learning procedures (as opposed to fixed bases models such as kernel methods), and thus this model class has been extensively studied in recent works to demonstrate the benefit of representation learning in neural networks trained with gradient descent -- see [AAM22] [BBSS22] and references therein. Note that such optimization-based advantage was not established in [GMMM20].
>
> As for why we do not consider the multiple-index setting, the main reason is that the optimization of neural network is less understood and more challenging to analyze. For instance, in order for the first gradient step to reveal information of the entire support of $f_*$, one may need to subtract the low-degree Hermite terms as done in [DLS22]. We therefore focus on the single-index target, which has been extensively studied in the isotropic setting (e.g., [ABJ21], [BES+22], [BBSS22], ...), and hence our results can be directly compared against existing benchmarks to illustrate the benefit of anisotropy in both KRR and NN+GD.
>
> **Q: Novelty of KRR analysis**
>
> We make the following remarks.
>
> 1. Our analysis of KRR builds upon previous results in [BMR21], [GMMM21], and [EK10]. However, our analysis is non-trivial as these prior works cannot be directly applied to KRR on an anisotropic Gaussian dataset with a divergent spike. The novelty of our proof comes from a combination of the proof techniques in [EK10, BMR21] and [GMMM21]. The former addresses the case when a nonlinear kernel with general data distribution is equivalent to a linear kernel; while the latter shows that a nonlinear kernel on unit sphere can be approximated by a polynomial kernel. In our case, we separate the data distribution into divergent spike and isotropic Gaussian; we then apply [EK10, BMR21] to show that the isotropic Gaussian part is equivalent to a linear kernel and will not help us to learn the target function; then, analogously to [GMMM21], we use the general Hermite polynomial expansion to exactly compute the performance of KRR on the spike direction. In the last step, since we are dealing with a polynomial kernel in one dimension similar to prior works, we omit some of the details by citing previous results. For completeness and convenience for readers, we will expand the proof and computations in our revision.
>
> 2. As the reviewer mentioned, the main difference between our kernel analysis and [GMMM20] is that we consider an anisotropic Gaussian dataset with one divergent spike in the covariance, whereas [GMMM20] considered a multi-index target with diverging number of directions. The proof strategies are also different: [GMMM20] directly used spherical harmonics to truncate the kernel due to the spherical setting; while in our setting, we need to first apply a rotation and a polynomial truncation to the kernel matrix to separate the isotropic Gaussian component and the spike.
> We note that our current approach cannot precisely characterize the performance of KRR under anisotropic Gaussian data with diverging number of spikes (as in [GMMM20] for spherical data), as we do not have a precise description of the orthogonal polynomial expansion for kernels under Gaussian data. We leave this direction as future work.
>
> **Q: Clarifications**
>
> * *lines 104-105:* We apply a normalization $\frac{1}{\sqrt{1+\theta}}$ in the teacher model to make sure that the order of the argument inside the link function is unchanged as we increase the signal-to-noise ratio $\theta$, and therefore the $L^2$ norm of $f_*$ also remains constant.
> * *line 165:* The "i.e." part is a sufficient condition of positive semidefiniteness for the kernel $k(x_i,x_j)=g(x_i^\top x_j)$. When $g$ is real analytic with nonnegative $g^{(k)}(0)$ for all $k\in\mathbb{N}$, we obtain a positive semidefinite inner product kernel.
> * *Scaling for plain GD/SGD:* In this work, we focus on the two-stage feature learning paradigm employed in recent works, including [AAM22], [BBSS22], [BES+22], [DLS22], and [ABAM23]; to our knowledge, we are the first to analyze the effectiveness of such procedure under anisotropic input data. Roughly speaking, we realize the representation learning advantage via gradient descent training of the first layer parameters, whereas the second-layer parameters can approximate the unknown link function.
> We believe that insights of our current results may be extended to NNs trained via standard GD, but this likely requires new proof techniques and a more involved analysis.
>
> **Q: Typos and corrections**
>
> Thank you for the close reading and detailed feedback. We have revised manuscript accordingly, and below we summarize a few noticeable corrections.
>
> * In the display of Theorem 3, we missed a square on the norm when stating the prediction risk of KRR.
>
> * Below line 258, there should be (i+1) factors in the sum since we consider the Hermite expansion of $\sigma'$.
>
> * In line 483, we need to exclude the case $p=0$ which corresponds to constant functions (that are easily learnable by KRR, e.g., see [BMR21]). In fact for simple presentation, in Corollary 4 we only consider teacher model with degree $p\ge 1$.
>
> * In line 506, the variance of the first coordinate is not equal to 1 and hence different from the other coordinates. We will use a different notation to distinguish between Hermite polynomial expansions with different variances (for the first coordinate vs. the remaining).
>
> We would be happy to clarify any concerns or answer any questions that may come up during the discussion period.

---

> > ### Comment · Reviewer_qX6D · 2023-08-19
> >
> > I thank the authors for their rebuttal. A few replies:
> >
> > **Motivation for the model.** I understand the motivations underlying the single-index model, but I was concerned by the relevance of studying simultaneously a single-index model and a spiked model. For this concern, a better reply was given to reviewer GruC. I encourage the authors to include this discussion in the paper, and, if possible, to support it with practical considerations. Note that this was a concern shared by three of the reviewers (including myself).
> >
> > **Novelty of the analyses.** I encourage the authors to add these discussions in the paper, for instance in an introduction of the appendix. This is usually very useful to articulate the different contributions in this line of work.
> >
> > **Clarifications.**
> > - l.165: I understand, but this deserves a proof or a reference, right?
> > - Scaling for plain GD/SGD: I understand the need of the two-step procedure for the theory, but it would still be interesting to try plain GD/SGD in experiments.

---

> > > ### Author Response · Authors · 2023-08-20
> > > **Reply to Reviewer qX6D**
> > >
> > > We thank the reviewer for the followup comment. Please find our response below.
> > >
> > > * **Additional discussions.** Thank you for the suggestion. We will include more discussions regarding the motivation of the studied model setting, as well as new ingredients in our theoretical analysis.
> > >
> > > * **Clarification.** The proof of the statement in l.165 can be found in Section 13.1 of [Scholkopf and Smola, 2002]. We will cite the reference and also provide a short proof for completeness.
> > > Schölkopf, Bernhard, and Alexander J. Smola. *Learning with kernels: support vector machines, regularization, optimization, and beyond*. MIT press, 2002.
> > >
> > > * **GD experiment.** Following your comment, we are currently running experiments for standard GD on the empirical loss. Preliminary results indicate that the findings are qualitatively similar, i.e., the sample complexity depends on the information exponent instead of the highest degree of $\sigma_*$. We will include this additional experiment in the updated Appendix.
> > >
> > > Please let us know if you have any additional questions.

---

> > > > ### Comment · Reviewer_qX6D · 2023-08-21
> > > >
> > > > Thank you for these clarifications and for running the additional experiments.

---

### Official Review · Reviewer_b5dX · 2023-07-04

**Soundness:** 3 good
**Presentation:** 3 good
**Contribution:** 2 fair
**Rating:** 6
**Confidence:** 3

**Summary:**

This paper considers the problem of learning a single-index target function under the spiked covariance data in the proportional asymptotic limit.
The results show the spike magnitude required for both kernel ridge regression (KRR) and two-layer neural networks (NN) trained with gradient descent and reveal that the NN can adapt to low-dimensional structures more effectively than KRR.
Numerical experiments on synthetic data are also supportive.

**Strengths:**

* The paper is clear and easy to follow.

* This paper shows theoretically the advantage of NN over KRR under a spiked convariance setting, which may provide more insight into the generalization of NN.

**Weaknesses:**

* It would be better to justify assumption of the single-index model considered in this paper.

* Although both upper and lower bounds for KRR are established, this paper only provides upper bound for NN.



**Questions:**

1. Can the results in this paper be applied to NN trained only by GD rather than the two-stage procedure considered in this paper?


**Limitations:**

The authors adequately addressed the limitations.

---

> ### Author Rebuttal · Authors · 2023-08-09
>
> Thank you for the comments and questions. We address the technical points below.
>
> **Q: Single-index assumption**
>
> The single-index model is a canonical setting of learning low-dimensional signals from high-dimensional data; it covers many statistical learning problems such as phase retrieval and generalized linear models. Importantly, the low dimensionality of the single-index target may be exploited by *adaptive* learning procedures (as opposed to fixed bases models such as kernel methods), and thus this model class has been extensively studied in recent works to demonstrate the advantage of representation learning in neural networks -- see [AAM22] [BBSS22] and references therein. We will include a more thorough discussion in the revised manuscript.
>
>
> **Q: Only upper bound for NN**
>
> Indeed we only provide an upper bound for the performance of NN+GD. This being said, for the purpose of establishing a separation between KRR and NN+GD, our upper bound (together with the sharp analysis of KRR) is already sufficient. Moreover, from the experiments in Section 5, we empirically observe that our derived upper bound is tight in terms of the required spike magnitude.
>
> Optimization lower bound for neural networks is generally difficult to establish. One possibility is to study the correlational statistical query (CSQ) lower bound, which relates to the gradient descent complexity (for the squared loss) -- see [DLS22] and [ABAM23] for the isotropic setting. We leave this direction as future work.
>
> **Q: Beyond the two-stage procedure**
>
> Our two-stage feature learning paradigm has been employed in many recent works, including [AAM22], [BBSS22], [BES+22], [DLS22], and [ABAM23], but to our knowledge, we are the first to analyze the effectiveness of such procedure under anisotropic input data. Roughly speaking, this two-stage process realizes the representation learning advantage via gradient descent training of the first layer parameters, whereas the second-layer parameters can approximate the unknown link function. We believe that insights of our current results may be extended to general NNs trained via standard GD, but this likely requires new proof techniques and a more involved analysis.
>
>
> We would be happy to clarify any concerns or answer any questions that may come up during the discussion period.

---

> ### Comment · Reviewer_b5dX · 2023-08-21
>
> Thanks for the response, but I will keep my score.

---

### Official Review · Reviewer_GruC · 2023-07-06

**Soundness:** 2 fair
**Presentation:** 3 good
**Contribution:** 3 good
**Rating:** 6
**Confidence:** 3

**Summary:**

The paper studies the learning capacity of Kernel Ridge Regression and 2-Layer Neural Network to approximate a target function $f^\star$ (denoted teacher) under a low dimensional structure. In particular, starting from a generative process where the covariance of the data has a low dimensional structure ($\mathbf{I}_d +\theta\mu\mu^\top$), the target function $f^\star$ depends on the signal $\mu$ and the objective is to approximate the target function as best as possible with a model known as a student. The authors investigate two models for the student in particular (Kernel Ridge Regression and 2-Layer Neural Network with one gradient step).

In these two models, the authors ask the question of the condition on the strength of the low dimensional structure ($\theta$) to approximate the function target $f^\star$. The conclusions of the study make it possible to highlight a scaling $\mathcal{O}(d^{1-1/d})$ and $\mathcal{O}(d^{1-1/k})$ respectively for the KRR and the NN where the link function of the target is a $p$-degree polynomial with Information exponent $k$, which makes it possible to highlight the ability of these two models to find the low dimensional structure and which also highlights the superiority in some cases of the NN. Experiments have been provided to corroborate the various theoretical conclusions.

**Strengths:**

1- Understanding the influence of low dimensional structures on the learning and generalization capabilities of machine models is an interesting question. The conclusions drawn by the authors seem to provide part of the answer although limited to a simple setting.

2- The paper is quite well written and easy to follow and although not having checked all the calculations, the theoretical results seem new and rigorous.

3- The experiments although in a synthetic setting corroborate the theoretical conclusions

**Weaknesses:**

1- The motivation for designing the target function $f^\star$ seems to be vague and has very little motivation. It is clear that we want to introduce a low dimension structure but why introduce it both in the covariance of the data and also as a weights (oracle) for the teacher network. One could choose weights that have nothing to do with the structure of the data and see how the structure of the data impacts the learning capacity. Also, the choice to put the low dimensional structure on the covariance matrix of the data deserves to be explained in accordance with the type of low dimensional structure generally encountered in real data. In particular the low dimensional structure can be found for example in the mean of the data in the form of sparsity, ...

2- The results obtained on the kernel seem to be a little too pessimistic or too loose. Indeed, one would have expected that the bounds depend on the type of kernel used and its proximity to the link function oracle $\sigma^\star$. Indeed, it is clear that for this problem where the means are zero, a linear kernel would give catastrophic performance compared to a quadratic kernel for example. I'm not sure that the boundary is tight enough. It is probably valid for the best choice of kernel. For example, I would be curious to see how Figure 2 and Figure 3 behave if instead of a Gaussian kernel one would take a quadratic kernel. Indeed, [1] proved that in the setting of zero mean data considered, the optimal kernel is not the Gaussian RBF but instead the quadratic kernel (see also other references in [2, 3]).

3- Experiments on real data would have been useful to illustrate the conclusions of the paper, for example by taking a pre-trained network as a teacher and learning a KRR and a 2-NN to approximate the teacher. The low dimension structure can be added manually or modelled.

References
[1] Covariance discriminative power of kernel clustering methods, A Kammoun, R Couillet - Electronic Journal of Statistics, 2023
[2] R. Couillet, F. Benaych-Georges, "Kernel Spectral Clustering of Large Dimensional Data", Electronic Journal of Statistics, vol. 10, no. 1, pp. 1393-1454, 2016
[3] H. Tiomoko Ali, A. Kammoun, R. Couillet, Random Matrix Asymptotics of Inner Product Spectral Clustering", IEEE International Conference on Acoustics, Speech and Signal Processing (ICASSP'18), Calgary, Canada, 2018.

**Questions:**

1- Can the authors motivate why they choose the convention for the low dimensional structure both for the data and the target function $f^\star$?
2- Can the authors comment on the independence of the result on the kernel used which has generally huge impact?
3- These conclusions are valid on real data as well or at least some insights on the actual performance for real data?

**Limitations:**

1- The main theoretical limitation is the lack of motivation of the choice of the teacher model. Why to model low dimensional structure as done by authors? Does this reflect the actual low dimensional structure encountered in real applications? The loose bounds (not depending on the kernel) also is a bit problematic

2- The main empirical limitation is the lack of insights for real world application, the lack of experimentation on different kernels, ...

---

> ### Author Rebuttal · Authors · 2023-08-09
>
> Thank you for the thoughtful feedback. We address the technical points below.
>
> **Q: low dimensional structure both for the data and the target function $f_*$**
>
> As remarked in the Introduction, the spiked covariance model is a classical statistical model that captures low-dimensional structure in high-dimensional data. The underlying intuition is that input directions with high variance are often good predictors of the labels -- indeed, this is the main reason that principal component analysis is used in practice.
> In this paper, we focus on the simplest spiked covariance setting with only one spike $\boldsymbol{\mu}\in\mathbb{R}^d$ and the target function $f_*$ is a *single-index model*. Note that this single-index setting has been extensively studied in many recent works as a canonical example of learning low-dimensional signals from high-dimensional data (e.g., [BAJ21] [BBSS22]), where the advantage of representation learning in neural networks can be demonstrated.
>
> In order for the spikes to have higher predictive power, the target function $f_*$ needs to align with these large directions -- such alignment between the features and target function is analogous to the *source condition* in the nonparametric literature.
> For this reason, we chose the single-index $f_*$ to be in the direction of $\boldsymbol{\mu}$, so that a larger spike corresponds to a stronger learning signal from the input data.
> In contrast, when the index features are orthogonal to spike direction, then we would not expect the low-dimensional structure to improve the learning performance: imagine the extreme setting where the spike completely dominates the remaining directions (i.e., the input is almost one-dimensional), then the learning problem becomes ``misspecified'' in the sense that the input features contain no information about the target function.
> As mentioned in the Conclusion section, an interesting future direction is to study the intermediate regime where $f_*$ is *partially* aligned with the spike direction.
>
>
>
> **Q: Pessimistic bounds for KRR**
>
> We emphasize that our characterization of KRR prediction risk is  *sharp* for inner-product kernels we consider, and does not depend on the particular choice of the kernel function, similar to [GMMM19] [GMMM20].
> The main difference between our results and the references [1][2][3] mentioned by the reviewer is that we allow the spectral norm of the covariance to *grow with the dimensionality $d$*. In this setting, when the spike magnitude is sufficiently large, the kernel matrix cannot be approximated via a low-degree Taylor expansion; consequently, it becomes possible for KRR to go beyond learning a low-degree $f_*$.
> More specifically, the references [1][2][3] considered spectral clustering by approximating the kernel as a linear kernel plus some low-rank spikes and using these low-rank spikes for clustering. These results does not handle the divergent spike in the dataset, and thus the kernel model cannot represent high-degree $f_*$. For instance, in the kernel clustering setting, [1] proved the optimal kernel is a quadratic kernel in classifying zero-mean datasets with different covariances; but in our case, it is clear that a quadratic kernel cannot achieve vanishing error when the target function is beyond degree-2.
> We will include a more detailed discussion in the revision.
>
> **Q: Insights on real-world data**
>
> While the focus of our work is mostly theoretical, the following high-level takeaways may transfer to more complicated and realistic datasets.
>
> 1. By introducing a low-dimensional structure to the input data in the form of a spiked covariance, both kernel ridge regression (KRR) and neural network trained by gradient descent (NN+GD) can achieve better performance in the high-dimensions. In other words, both methods can benefit from the anisotropy of input data, provided that the target function (ground truth) focuses on the dominant feature directions.
>
> 2. The performance of KRR and NN+GD depends on different notions of target complexity: for KRR the complexity of learning is measured by the *degree* of $f_*$, whereas for NN+GD it is the *information exponent*, which determines the difficulty of gradient-based optimization (note that this differs from prior results such as [GMMM20], where the optimization complexity of NN is not studied). Since the information exponent is by definition no larger than the degree of $f_*$, we know that NN+GD can adapt to low-dimensional structures more effectively than KRR.
>
> We would be happy to clarify any concerns or answer any questions that may come up during the discussion period.

---

### Official Review · Reviewer_sXLf · 2023-07-08

**Soundness:** 3 good
**Presentation:** 3 good
**Contribution:** 3 good
**Rating:** 5
**Confidence:** 4

**Summary:**

This paper studies a problem of parameter recovery under a generalized linear model-like setting. The novelty of this problem is revisiting this setting in the proportional limit setting: The sample size divided by the feature dimension converges to a fixed constant as both increases to infinity. This is a high-dimensional setting that received interest in the statistics theory community for many machine learning problems. This paper aims to study this setting with two learning methods: The kernel ridge regression and two-layer neural networks trained by gradient descent.

For both settings, the paper contributes tight bounds on recovering the unknown signal vector, given feature vector and label vectors in the proportional limit setting. Empirical studies validate the theoretical results.



**Strengths:**

S1. A solid paper with strong technical contributions. Working out the asymptotic rates for kernel ridge regression and two-layer neural nets is a highly nontrivial problem, and the results shed some light on when one is better than the other, depending on the relation between signal and dimension.

S2. The paper is overall nicely written. The discussion with respect to related work is also clear to read.

**Weaknesses:**

W1. Although the results are strong, there is a lack of insight that one can take away from the analysis. I consider this a limitation because such discussions have been commonly stated in existing random matrix theory / machine learning papers, e.g.,

Hastie, T., Montanari, A., Rosset, S., & Tibshirani, R. J. (2022). Surprises in high-dimensional ridgeless least squares interpolation. Annals of statistics, 50(2), 949.

Wei, A., Hu, W., & Steinhardt, J. (2022, June). More than a toy: Random matrix models predict how real-world neural representations generalize. In International Conference on Machine Learning (pp. 23549-23588). PMLR.

**Questions:**

- Would the result hold for anisotropic input data?
- Does the result apply to classification problems?
- What if $y$ is contaminated by some random noise (which is a common assumption for this type of setting), or is it the case that the results already apply to this case?

**Limitations:**

Limitations (and extensions) have been discussed in the conclusion section.

Potential negative societal impact would be unexpected given the technical nature of the paper.

---

> ### Author Rebuttal · Authors · 2023-08-09
>
> Thank you for the comments and questions. We address the technical points below.
>
> **Q: Lack of insight**
>
> Here we briefly summarize the takeaways of our analysis.
>
> 1. By introducing a low-dimensional structure to the input data in the form of a spiked covariance (which is a common model in high-dimensional statistics), both kernel ridge regression (KRR) and neural network trained by gradient descent (NN+GD) can achieve better performance in the high-dimensional asymptotic limit.
> In other words, both methods can benefit from the anisotropy of input data, provided that the target function (ground truth) focuses on the dominant feature directions.
>
> 2. Similar to the isotropic setting, the performance of KRR and NN+GD depends on different notions of target complexity: for KRR the complexity of learning is measured by the *degree* of $f_*$, whereas for NN+GD it is the *information exponent*, which determines the difficulty of gradient-based optimization (note that this differs from prior results such as [GMMM20], where the optimization complexity of NN is not studied).
> Since the information exponent is by definition no larger than the degree of $f_*$, our analysis indicates that NN+GD can adapt to low-dimensional structures more effectively than KRR.
> These theoretical predictions are supported by experiments.
>
>
> **Q: Anisotropic input data**
>
> Note that the spiked covariance input data we consider is by definition anisotropic. We, therefore, interpret the reviewer's comment as the extension to *arbitrary* covariance, which our current analysis does not cover. As remarked in the Introduction, most existing results on this topic only handled the isotropic setting, whereas we show that the addition of one single spike (arguably the simplest anisotropic setting) already gives rise to
> rich phenomenologies, such as learnability of a wider range of target functions in the *proportional regime*, and a new separation between the performance of KRR and NN+GD. We believe that our analysis serves as an important first step toward the understanding of learning behavior under more complicated and realistic structured data.
>
>
> **Q: Extension to classification problems**
>
> In this submission we focus on the regression setting, for which the performance of KRR and NN has been extensively studied under high-dimensional isotropic input data (see Introduction); this provides a useful benchmark for us to compare our findings against (e.g., the difficulty of gradient-based learning measured by the information exponent).
> It is definitely interesting to extend our current analysis to the classification setting, where the spike structure can be encoded in different ways, such as a Gaussian mixture model.
>
> **Q: Handling of label noise**
>
> Recall that the prediction risk admits a bias-variance decomposition, where the bias term comes from the learning of target function $f_*$, and the variance term is due to contamination of label noise $\varepsilon$. Our current analysis focuses on the bias term (i.e., the noiseless setting), which is more challenging to characterize. When i.i.d. label noise is added, it can be shown via standard arguments that under an appropriate choice of regularization $\lambda$, the variance term in both KRR and NN+GD vanishes when $n/d\to\psi$ becomes large.
> We will include a discussion on the handling of label noise in the revised manuscript.
>
> We would be happy to clarify any concerns or answer any questions that may come up during the discussion period.

---

> > ### Author Response · Authors · 2023-08-20
> > **Follow up**
> >
> > Dear reviewer, as the end of the discussion period is approaching, we wanted to follow up on our rebuttal and make sure that we have addressed your concerns. Please let us know if you have any additional questions.

---

### Decision · Program_Chairs · 2023-09-21

**Decision:**

Accept (poster)

**Comment:**

In this paper, the authors study the problem of learning a scalar valued target function when data is provided with a spiked covariance structure.  The authors provide theoretical justification of the spike size necessary for detection by Kernel Ridge Regression and a two-layer neural network trained using gradient descent.  The results show that the performance of these methods depends on different notions of complexity: one depends on the degree the function and the other on its information exponent.  The strength of the paper is the novel theoretical results that identify relevant notions of complexity under which learning guarantees can be proved under data with a particular spiked covariance structure.  The results notably provide qualitative insight that neural networks can adapt to low-dimensional structures more effectively than Kernel Ridge Regression can under this structured data.  Several reviewers suggested that the paper could be improved by better justification of the single-index model being considered.